ecology, environmental science, behaviour

heat dissipation limit theory, climate change, equatorial, thermoregulation, banded mongoose, cooperative breeder

**Author for correspondence:**
Hazel J. Nichols
e-mail: h.j.nichols@swansea.ac.uk

# Small increases in ambient temperature reduce offspring body mass in an equatorial mammal

Monil Khera[1], Kevin Arbuckle[1], Francis Mwanguhya[2], Solomon Kyabulima[2], Kenneth Mwesige[2], Robert Businge[2], Jonathan D. Blount[3], Michael A. Cant[3] and Hazel J. Nichols[1]

[1]Department of Biosciences, Swansea University, Swansea SA2 8PP, UK
[2]Banded Mongoose Research Project, Queen Elizabeth National Park, Kasese District, Uganda
[3]Centre for Ecology and Conservation, University of Exeter, Cornwall, UK

KA, 0000-0002-9171-5874; JDB, 0000-0002-0016-0130; MAC, 0000-0002-1530-3077; HJN, 0000-0002-4455-6065

Human-induced climate change is leading to temperature rises, along with increases in the frequency and intensity of heatwaves. Many animals respond to high temperatures through behavioural thermoregulation, for example by resting in the shade, but this may impose opportunity costs by reducing foraging time (therefore energy supply), and so may be most effective when food is abundant. However, the heat dissipation limit (HDL) theory proposes that even when energy supply is plentiful, high temperatures can still have negative effects. This is because dissipating excess heat becomes harder, which limits processes that generate heat such as lactation. We tested predictions from HDL on a wild, equatorial population of banded mongooses (*Mungos mungo*). In support of the HDL theory, higher ambient temperatures led to lighter pups, and increasing food availability made little difference to pup weight under hotter conditions. This suggests that direct physiological constraints rather than opportunity costs of behavioural thermoregulation explain the negative impact of high temperatures on pup growth. Our results indicate that climate change may be particularly important for equatorial species, which often experience high temperatures year-round so cannot time reproduction to coincide with cooler conditions.

## 1. Background

Human-induced climate change is causing average temperatures to rise and is increasing the frequency, severity and duration of heatwaves [1]. High temperatures have been shown to negatively impact a variety of key mammalian life-history traits including body size, reproductive success and survival, which may in turn lead to population declines [2–5]. Many species attempt to avoid hyperthermia by changing their behaviour, for example through a reduction in activity or microhabitat selection of cooler locations [6,7]. However, these strategies, collectively called 'behavioural thermoregulation', can be costly in terms of lost opportunities because they require animals to alter their patterns of behaviour, which can carry significant fitness costs. For example, animals may cease foraging during hot periods, which can reduce their energy intake [8].

High temperatures may also be costly, even when there is ample food available from the environment [9]. The heat dissipation limit (HDL) theory proposed by Speakman *et al.* [10] suggests that that as air temperature increases, it becomes harder for metabolic heat to be lost, and if heat is generated faster than it can be lost, this will cause hyperthermia. The reduced capacity for heat to be dissipated

(rather than energy availability) may therefore be the primary limiter of maximum energy expenditure [10].

Lactation is considered the most energetically costly event in a female mammal's lifetime [11], and milk production is a highly exothermic process [12,13], leading to lactating animals being particularly vulnerable to chronic hyperthermia [14]. In support of the HDL theory, experimental studies on captive mice (*Mus musculus*) [12,15], pigs (*Sus scrofa domesticus*) [16], dairy cattle (*Bos taurus*) [17] and hamsters (*Cricetulus barabensis*) [18] have found high temperatures to depress milk production and reduce offspring growth, although it is unclear from these studies whether females stop lactating before or after mild hyperthermia sets in. The HDL theory has also been supported by multiple fur-removal experiments [11,19–21], whereby removing fur increases thermal conductance, allowing for greater heat dissipating capacity.

So far, the HDL theory has been predominantly tested via laboratory experiments, with very few studies testing the HDL theory in the wild (but see Nilsson *et al.* [22] who studied HDL in relation to brood care in birds, rather than lactation in mammals). When studies have been conducted in captivity, food is given ad libitum, such that the HDL theory in relation to lactation remains untested under limited food resources. In natural systems, food supply largely determines energy availability and can have a strong impact on reproductive output [23], so studies of wild systems are required to understand the relative importance of high temperatures and food supply on lactation and offspring growth [23].

There is also a lack of studies investigating the impact of high temperatures in equatorial species. Animals living close to the equator at low altitudes experience relatively high temperatures year-round with little seasonal variation compared to those in temperate regions. Equatorial species are therefore likely to be physiologically adapted to relatively constant temperatures, and so may have narrow thermal ranges which could leave them particularly susceptible to even small changes in temperature [24,25]. Furthermore, in temperate regions, high seasonal temperatures are generally associated with an increase in food availability [26] which can make it difficult to distinguish between the direct effect of temperature on reproductive output versus indirect effects via impacts on food supply [23]. Tropical regions, however, are characterized by high seasonal variation in rainfall, which is often the main determinant of food supply [27–30]. Studying equatorial species can therefore allow us to decouple the impacts of temperature variation and food supply on energy dynamics.

Here, we test the HDL theory in a wild population of banded mongooses (*Mungos mungo*) in equatorial Uganda. Banded mongooses live in family groups where females (mean = 3.5 females, range 1 to 13) give birth synchronously (usually on the same night) to between one and five pups each [31]. The resultant litters are raised communally in an underground den, and pups are suckled indiscriminately by multiple lactating females for approximately 30 days before the weaning process begins [31,32]. Underground rearing is likely to buffer pups against direct negative effects of high temperatures [33], therefore separating thermal effects on pups from those on lactating females. This makes them ideal for studying the indirect effects of high temperatures on pup growth via its effect on milk production. Furthermore, banded mongoose adults behaviourally thermoregulate by foraging when temperatures are cooler, resting in the shade during the hottest parts of the day [34]. High temperatures therefore likely result in reduced time available for foraging [35]. However, rainfall is tightly linked to the availability of invertebrate prey [36–38], allowing us to investigate whether high food availability can compensate for the negative impacts of high temperatures.

## 2. Methods

Our study used life history, body mass, genetic and environmental data collected between August 2000 and March 2018 from a population of wild banded mongooses in Queen Elizabeth National Park, Uganda (0°12′ S, 27°54′ E). At any one time, the population was made up of 8–12 social groups, each of which typically contained between 10 and 30 adults [39]. Banded mongooses primarily fed on invertebrates [40] and while groups foraged together, foraging itself was not cooperative [41]. Groups undertook two foraging sessions per day; the first session beginning shortly after dawn and usually lasting between 3–4 h, and the second session beginning in the afternoon, usually lasting 2–3 h and finishing before sunset [34,35].

### (a) Climatic variables

Our study site is located in a scrub–savannah habitat and can be characterized by relatively constant temperatures (monthly mean maximum daily temperature ± s.d. = 29.5 ± 1.5°C) [42]. Nevertheless, short-term variation does occur, including heatwaves [43]. There are also two rainy seasons per year, from March–May and August–December, with drier periods in between. We used rainfall as a proxy for food availability as invertebrate prey are more abundant at higher rainfall [36,37], and rainfall has previously been shown to positively affect weight gain in adults [42] and pups [44]. Data on rainfall (mm) and maximum ambient temperature (Tmax) (°C), both measured to one decimal place, were collected daily from a weather station located in our study site.

### (b) Life history

Our study population is habituated to observation at less than 10 m (usually less than 5 m) [45]. Each social group was visited every 1–3 days to determine group composition and record births, deaths and other life-history data [46]. Pregnancies and births were identified by changes in the size and shape of the abdomen, the absence of previously pregnant females on foraging trips the morning after birth, and the start of pup-care behaviour [31,39,47].

### (c) Body mass

Since pups are raised in underground dens it was not possible to weigh them until they emerged at approximately 30 days old, after which they start accompanying the group on foraging trips and wean onto solid food [39]. Pups were caught by hand, sexed and weighed using a portable electronic scale (accuracy ± 1.5 g) in the morning (*ca* 07.30 am) (see Jordan *et al.* [45]). Due to time constraints and field researcher safety considerations, there was some variation in the age of pups at first weighing. This study included 215 pups weighed when they were between 28 and 38 days old (mean mass = 188.9 g (range 87–307 g); mean age at weighing = 32.9 days), which captures an age range at which weighing pups is possible but when pups still rely heavily on milk.

### (d) Maternity

Synchronized birthing masks the maternity of banded mongoose pups so maternity cannot be determined behaviourally. Instead,

royalsocietypublishing.org/journal/rsbl   Biol. Lett. **19**: 20230328

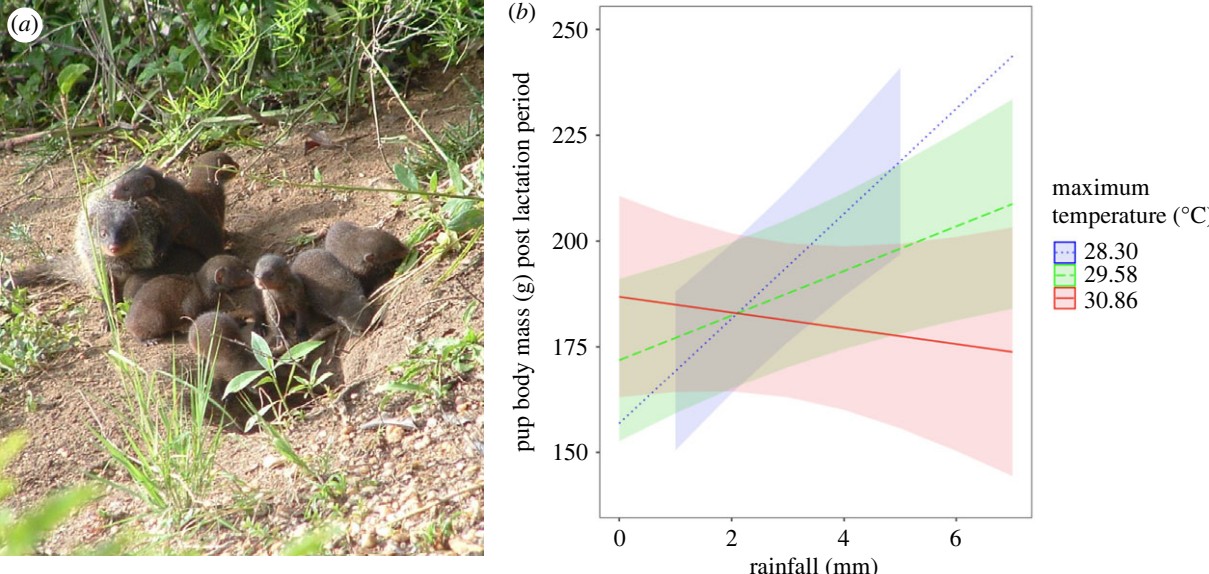

**Figure 1.** (a) Banded mongoose pups emerging from their underground den at around 30 days old. (b) Pup body mass as a function of the interaction between temperature and rainfall. Lines show the predictions from the LMM, plotted for three temperatures; 28.3 (mean − 1 s.d.), 29.6 (mean), and 30.9°C (mean + 1 s.d.), with the shaded areas representing 95% CI. Temperature was included as a continuous variable in our analysis, but we show the predicted rainfall–pup mass relationship at three temperatures for illustrative purposes. At low and medium temperatures, pup weight increases with rainfall, however at high temperatures, there is no effect of rainfall on pup weight (as evident from the CIs).

**Table 1.** Summary of an LMM investigating pup mass after the lactation period. Our model included 215 pups, with 60 different mothers, born into nine social groups.

| fixed effects | estimate | se | d.f. | t-value | p-value |
|---|---|---|---|---|---|
| intercept | −269.983 | 135.505 | 191 | −1.99 | 0.048 |
| age | 3.132 | 0.884 | 198 | 3.55 | $4.90 \times 10^{-4}$ |
| sex (male) | 5.986 | 4.506 | 182 | 1.33 | 0.186 |
| number of lactating females | −1.598 | 1.343 | 194 | −1.19 | 0.235 |
| rainfall | 170.137 | 42.966 | 188 | | |
| temperature | 11.712 | 4.649 | 191 | | |
| temperature : rainfall | −5.574 | 1.458 | 187 | −3.82 | $1.79 \times 10^{-4}$ |

maternity data were extracted from an existing genetic pedigree based on 35–43 microsatellite loci; see [48] and [49].

## (e) Statistical analysis

We constructed a linear mixed effects model (LMM) in R 3.3.1 [50], with pup mass as our response variable. To investigate how rainfall affected pup mass at varying temperatures, we modelled the interaction between rainfall and temperature. We used mean daily Tmax (°C) during the lactation period (0–30 days) and the mean rainfall (mm) over the 30 days prior to the birth of the pup birth as a proxy for food availability during the lactation period, since it takes this time for high rainfall to translate into higher food availability [36–38]. Previous studies also indicate that rainfall over this time period is positively correlated with adult daily weight gain [42]. In our model, mean Tmax ranged from 27.0–32.1°C and mean rainfall values ranged from 0.1 to 6.5 mm. Our rainfall and temperature measures were not strongly correlated (correlation coefficient = 0.081). The number of lactating females present in the group was added as a covariate since pups with access to more lactating females may be able to obtain more milk, although these pups will likely also face higher competition from other pups. Sex was fitted as a covariate since male pups were previously found to weigh slightly more

than females [51], and pup age was included to account for age differences in weighing. It is possible that offspring weight might vary by mother [52] so we fitted the identity of the mother as a random effect. We also fitted the identity of the social group as a random effect to account for variation in group-level factors such as territory quality. Standard model checks were employed following Crawley [53].

## 3. Results

Pup mass was significantly affected by the interaction between temperature and rainfall (table 1). Under cooler temperatures, higher levels of rainfall were associated with heavier pups, however, under hotter conditions, changes in rainfall had little effect on pup mass (figure 1). Our model controlled for the significant effect of pup age at weighing, and we found no effect of pup sex or the number of lactating females.

## 4. Discussion

We found that under hotter conditions, increased food availability did not lead to increased banded mongoose pup

mass. This is consistent with the HDL theory, which proposes that as the air temperature gets closer to body temperature, dissipating heat becomes harder [10]. As a result, lactating females may be forced to suppress exothermic processes such as milk production (either at or approaching their critical thermal maximum) in order to avoid hyperthermia [15]. At lower ambient temperatures, however, banded mongoose pup body mass increased with food supply. Here, heat dissipation can happen faster, which may lift constraints on how quickly energy can be metabolized [10]. As a result, when there is sufficient food available, females can consume more energy to increase milk production, which is consistent with our finding of greater pup growth.

In addition to placing constraints on lactation, high temperatures could also directly affect the pups' ability to dissipate heat causing them to reduce milk intake in order to avoid hyperthermia, although likely to a lesser extent than adults due to the pups' higher surface area to volume ratios [23]. Similar effects have been found in meerkats, whereby weaned pups have reduced mass gain at high temperatures, but without an apparent reduction in feeding rate [54]. However, in our study of banded mongooses, we focused on pups that are raised in underground dens pre-weaning. While there have been no studies of the thermal properties of banded mongoose dens, similar structures have been shown to provide insulation against temperature fluctuations in a variety of other species [55–57]. Dens are therefore likely to provide protection to banded mongoose pups against direct negative effects of high temperatures. Lactating females, however, must forage in ambient temperatures, thus limitations on lactation are likely to produce the greatest impacts on pup growth in this species.

High temperatures during the lactation period are likely to have downstream impacts on pups post-weaning. For example, pups that are lighter at emergence from the den are less likely to survive to nutritional independence (90 days) [44,58] and weigh less at maturity (1 year) [58]. Furthermore, body mass at maturity influences lifetime reproductive success, with lighter individuals of both sexes producing fewer pups [58]. Hot conditions experienced in early life could therefore have lifelong fitness implications.

High temperatures are also likely to have impacts that go beyond body mass. For example, high seasonal temperatures have been shown to reduce the probability of the communal litter surviving to 30 days [35]. As our findings are based on those pups that survived the lactation period, we may have missed cases where lactation has been reduced to the extent that it has caused pup mortality prior to the emergence of the litter from the den. Unfortunately, it is rarely possible to observe or weigh banded mongoose pups while in the den, so causes of pre-emergence mortality are difficult to determine.

High temperatures during the early developmental stages have also been found to negatively impact other species. For example hot conditions reduce the survival of southern pied babblers (*Turdoides bicolor*) to independence [5], reduce mass gain and fledgling mass in common fiscal nestlings (*Lanius collaris*) [59] and reduce mass gain and weight at nutritional independence in meerkats [54]. Reduced body size is in turn associated with reduced survival, fecundity and reproductive success [60–62]. Therefore, the negative impacts of hot conditions during development on lifelong fitness could be relatively common among birds and mammals, although the importance of HDL across species is currently unclear and needs testing.

As global temperatures rise and heatwaves increase in frequency and intensity [1], temperate species may be able to adjust their reproductive phenology [63,64], for example to avoid lactating when seasonal temperatures are high. It is unclear, however, if and how equatorial species will be able to adapt to these changes. Our study confirms the susceptibility of an equatorial species to small changes in temperature; mean daily maximum temperatures over the lactation period only ranged from 27.0 to 32.1°C. Our results also imply that high food abundance may not compensate for the negative impacts of high temperatures on lactating females. Although rainfall is predicted to increase in western Uganda with climate change [65], temperatures across Uganda are also continuing to rise [66] and so over time, increased rainfall may not compensate for higher temperatures, and rainfall may cease to predict pup mass. In the light of rising global temperatures as well as more intense and frequent heatwaves [1], we highlight a clear need for greater research efforts on the effect of climatic variation on species occupying tropical and equatorial regions, where populations live and breed under consistently high temperatures.

**Ethics.** This study was approved by the Ethical Review Committees of the Universities of Exeter and Swansea (010323/4401), the Uganda Wildlife Authority (COD/96/05) and Uganda National Council for Science and Technology (NS443ES), and adhered to the Guidelines for the Treatment of Animals in Behavioural Research and Teaching, published by the Association for the Study of Animal Behaviour.

**Data accessibility.** Data are available from the Dryad Digital Repository: https://doi.org/10.5061/dryad.3bk3j9krw [67].

Data available as part of the electronic supplementary material [68].

**Declaration of AI use.** We have not used AI-assisted technologies in creating this article.

**Authors' contributions.** M.K.: conceptualization, data curation, formal analysis, funding acquisition, investigation, methodology, project administration, validation, visualization, writing—original draft, writing—review and editing; K.A.: conceptualization, methodology, project administration, supervision, writing—review and editing; F.M.: data curation, investigation, methodology, project administration, resources, writing—review and editing; S.K.: data curation, investigation, methodology, project administration, writing—review and editing; K.M.: data curation, investigation, writing—review and editing; R.B.: data curation, investigation, writing—review and editing; J.D.B.: funding acquisition, writing—review and editing; M.A.C.: data curation, funding acquisition, project administration, resources, writing—review and editing; H.J.N.: conceptualization, methodology, project administration, supervision, writing—original draft, writing—review and editing.

All authors gave final approval for publication and agreed to be held accountable for the work performed therein.

**Conflict of interest declaration.** We declare we have no competing interests.

**Funding.** This work was supported by the Natural Environment Research Council (grant no. NE/N011171/1) and the European Research Council (grant no. 309249). H.J.N. was supported by an Alexander von Humboldt Foundation Research Fellowship.

**Acknowledgements.** We thank Uganda Wildlife Authority and Uganda Council for Science and Technology for permission to conduct our research, and the wardens of Queen Elizabeth National Park for support with our long-term study. We are very grateful to the Uganda field team, and past and present researchers for the long-term data collection. We also thank David Wells for his help and advice on generating the pedigree.

royalsocietypublishing.org/journal/rsbl Biol. Lett. 19: 20230328

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
