## [Peer Review File · Biology Letters]

Review History

RSBL- 2023-0328.R0 (Original submission)

Review form: Reviewer 1

Recommendation

Accept with minor revision (please list in comments)

Scientific importance: Is the manuscript an original and important contribution to its field?

Excellent

General interest: Is the paper of sufficient general interest?

Excellent

Quality of the paper: Is the overall quality of the paper suitable?

Excellent

Is the length of the paper justified?

Yes

Should the paper be seen by a specialist statistical reviewer?

No

Do you have any concerns about statistical analyses in this paper? If so, please specify them explicitly in your report.

No

It is a condition of publication that authors make their supporting data, code and materials available - either as supplementary material or hosted in an external repository. Please rate, if applicable, the supporting data on the following criteria.

Is it accessible?

Yes

Is it clear?

Yes

Is it adequate?

Yes

Do you have any ethical concerns with this paper?

No

Comments to the Author

This manuscript, "Small increases in temperature reduce offspring mass in an equatorial mammal" is exceptionally well thought out, designed, written and reasoned. I have only a two minor edits and one suggestion.

1) on line 162: "at after" the "at" could be deleted, since it is redundant.

2) line 189: "pups" should be "pups'" since this is a possessive word as written.

My suggestion is that the concept of a heat dissipation limit appeared a number of years before papers by Speakman and Krol. The authors may want to check out a paper by Porter et al. (Porter et al. 1994). Figures 15, 22 and 23 in that paper casts thermodynamic constraints on energetics due to climate in a much broader context, including the material presented in this manuscript.

Porter, W., J. Munger, W. Stewart, S. Budaraju, and J. Jaeger. 1994. Endotherm Energetics - From a Scalable Individual-Based Model to Ecological Applications. Australian Journal of Zoology 42:125-162.

Review form: Reviewer 2

Recommendation

Major revision is needed (please make suggestions in comments)

Scientific importance: Is the manuscript an original and important contribution to its field?

Good

General interest: Is the paper of sufficient general interest?

Good

Quality of the paper: Is the overall quality of the paper suitable?

Poor

Is the length of the paper justified?

Yes

Should the paper be seen by a specialist statistical reviewer?

No

Do you have any concerns about statistical analyses in this paper? If so, please specify them explicitly in your report.

Yes

It is a condition of publication that authors make their supporting data, code and materials available - either as supplementary material or hosted in an external repository. Please rate, if applicable, the supporting data on the following criteria.

Is it accessible?

Yes

Is it clear?

Yes

Is it adequate?

No

Do you have any ethical concerns with this paper?

No

Comments to the Author

You have a special, and perhaps unique, data set of body masses, at emergence from the den, of 215 banded mongoose pups, all of which had experienced similar, benign, thermal environments in the den, and all of which had had similar access to food ("pups are suckled indiscriminately by multiple lactating females", line 84). Mean mass at weighing was 189 g but body masses at emergence varied widely, from 93g to 307 g (supplemental information) or 113% of mean mass. Age at weighing varied by about 10 days and age was associated very powerfully with mass (Table 1). Assuming that other factors in the den that could have affected mass at emergence (e.g. incidence of disease, predation) were similar for all pups, it is plausible that differences in mass were associated with the amount of milk delivered by the multiple lactating females. It also is plausible that the amount of milk delivered was associated with the weather during lactation, which would have affected both the thermoregulation of the lactating females but also the amount of food available to them, because the abundance of (predominantly invertebrate) prey increased with increasing rainfall.

It is whether pup mass was associated with the weather to which the lactating females were exposed that you set out to analyse. You concluded, using LMM analysis, that, on their own, neither the maximum air temperature over the period of lactation nor the rainfall at a time earlier, to take account of latency in consequences for prey abundance, had any significant effect on pup mass, but the interaction between maximum temperature and earlier rainfall indeed did so.

Unfortunately, in my opinion, you were so committed to finding support for the heat dissipation limit (HDL) theory of Speakman and Król (reference #9) that you failed to address the data before you on its merits and you overlooked many deficiencies in the arguments that you used to reach your conclusions. In my view, I do not believe there was any solid evidence provided for factors contributing to variability in pup mass at emergence other than age. Also in my view, HDL theory warranted no more than a paragraph in the "discussion".

General comments

1. Animals undoubtedly can be exposed to environments in which they have difficulty dissipating body heat. That's simply a consequence of the laws of physics, not a consequence of HDL theory. Those environments typically will have high radiant heat load and high water-vapour pressure, inhibiting evaporative cooling. There is no information about either radiation or water-vapour pressure in your paper. In such environments, exothermic processes may well cause detrimental hyperthermia (not all hyperthermia is detrimental), and that hyperthermia may reduce or eliminate the exothermic process. Runners who get too hot have to stop running. That's a feedback process. What's different about HDL theory is that it proposes a feedforward process. HDL proposes that animals, not yet hyperthermic, receive some kind of warning signal about impending thermal threats and reduce exothermic processes in anticipation of running into trouble. So, to claim that reducing an exothermic process like lactation is consistent with HDL, you have to demonstrate that the reduction is anticipatory, a demonstration for which I don't think that your data are suitable. A study design that could test the theory might be one in which animals were under the same ambient thermal load in two situations, in one where ambient heat loads were in the process of increasing (threat coming) and in another where ambient heat loads were in the process of decreasing (no threat coming). Then, according to HDL theory, animals would reduce exothermic processes in the first situation but not the second. So, I don't think that you should construct your paper around HDL.

2. It has been known for at least 70 years that high ambient heat load compromises the physiology of lactation, potentially resulting in massive reduction of milk production, especially in high-lactating animals. See the attached diagram for cows, from Hafez, E.S.E., 1968. Adaptation of domestic animals, based on 1950 data. (See Appendix A)
If your lactating females were under high enough heat load, their milk production may well have been reduced without them undertaking any anticipatory action to reduce exothermic processes. We do not know whether they were under high heat load because the important source of heat would have been solar radiation, which was not reported.

3. Your crucial results are presented in Table 1 and Figure 1. I believe the regression lines of Figure 1, drawn without any data points, conceal some problematic properties of your data set. For example, from the supplemental information, when rain was more than 4 mm, there were only four pups for which lactating females were exposed to maximum air temperature less than 28°C but ten for which lactating females were exposed to more than 32°C. When rain was less than 2 mm, there were 17 pups for which lactating females were exposed to maximum air temperature less than 28°C but none for which lactating females were exposed to more than 32°C. So the pups were not distributed anywhere near evenly across combinations of rainfall and maximum air temperature to which the lactating females were exposed, as valid regression analysis requires, though the figure may give the impression that they were. Further, again from the supplemental information, the mass of pups for which the lactating females were exposed to rain less than 2 mm and maximum air temperature less than 28°C was 155 ± 34 (SD) g; according to your formulation those females would have had the least food intake and been able to dissipate metabolic heat most easily. The mass of pups for which the lactating females were exposed to rain more than 4 mm and maximum air temperature more than 32°C was 149 ± 29 g; according to your formulation those females would have had the highest food intake and been able to dissipate metabolic heat least easily. But a t-test shows that those mean masses were not significantly different. So the two most-extreme combinations of temperature and rainfall produced pups of the same mass, which is hard to reconcile with your conclusion that the temperature/rain interaction was the weather factor with which changes in mass were associated.

4. The single measure of the status of the thermal environment that you used was the maximum air temperature over the period of exposure. But you point out (lines 106-109) that the lactating females were not active in the middle of the day, which means that they were not exposed to those maximum temperatures. The appropriate temperature to enter into your analysis, I believe,

would be the temperature during the foraging bouts, say temperature for the three hours after dawn and the three hours before dusk, averaged over all the days of the exposure.

5. Were rainfall and maximum air temperature independent variables? I'd like to see a plot of rainfall against maximum air temperature. Also, one way in which rainfall and ambient heat load on the animals could have interacted would be if cloud cover increased in the rainier months and so reduced solar radiation. Without a measure of radiation, you can't test that possibility directly. However, I believe that you should re-run your LMM with season as a covariate. Since conventional seasons aren't relevant at the equator, I think that you should designate four seasons around rainfall, for example Season 1 = March-May, Season 2 = June-August, Season 3 = September-November, Season 4 = December- February. That re-run also should use average temperature at the time of foraging and not maximum air temperature.

6. You say (lines 79-80), I believe correctly, that "Studying equatorial species can therefore allow us to decouple the impacts of temperature variation and food supply on energy dynamics". However, studying equatorial species makes it very difficult to discover effects resulting from differences in the thermal status of the environment, because that thermal status varies so much less on the equator than it does at higher latitudes. One of the reasons for temperature not turning up as a factor on its own in your LMM might be that there just was not enough variation in temperature.

7. You have ignored evaporative cooling. Evaporative cooling is by far the most-effective avenue for dissipating body heat, and, except when ambient water-vapour pressure is too high, can dissipate the heat produced by strongly exothermic processes, including heavy exercise. Evaporative cooling certainly can dissipate, easily, the heat produced by lactation, and sustained evaporative cooling is perfectly possible in equatorial environments where water is readily available. I am aware that banded mongoose are reluctant drinkers but there will be plenty of water available in their prey. I do not know of measurements of evaporative cooling in banded mongoose, but it has been described, and is powerful, in slender mongoose, the range of which overlaps with that of banded mongoose (Kamau, J.M., Johansen, K. and Maloiy, G.M.O., 1979. Thermoregulation and standard metabolism of the slender mongoose (*Herpestes sanguineus*). *Physiological Zoology*, 52(4), pp.594-602), as well as in suricates (Müller, E.F. and Lojewski, U., 1986. Thermoregulation in the meerkat (*Suricata suricatta* Schreber, 1776). *Comparative Biochemistry and physiology. A, Comparative Physiology*, 83(2), pp.217-224). If the banded mongoose has similar evaporatively cooling capacity, and provided ambient water vapour pressure was not too high, your lactating females would have no difficulty dissipating body heat, including heat produced during lactation, even at your hottest air temperature, and even under high solar radiation..

Line by line comments

8. Line 28: equatorial species experience similar temperatures year-round, but those living at high altitude on the equator do not experience high temperatures year-round.

9. Line 44: there are many other strategies that fall under the rubric of "behavioural thermoregulation".

10. Line 45: ceasing foraging in hot periods does not necessarily reduce energy intake. See Hetem, R.S. et al., 2012. Activity re-assignment and microclimate selection of free-living Arabian oryx: responses that could minimise the effects of climate change on homeostasis? *Zoology*, 115(6), pp.411-416.

11. Line 71: again, not at high altitude.

12. Line 75: you need to make clear that, in temperate environments, increasing temperature is associated with increasing food, because in arid environments, increasing temperature is associated with less food.
13. Line 81: against what alternative or alternatives did you test HDL theory? I do not believe that you were testing the theory. You were seeking support for it.
14. Line 86: underground rearing buffers the pups much more against the effects of solar radiation than it buffers then against high air temperature.
15. Line 87: what underground rearing makes possible is separating thermal effects on pups (see van de Ven et al., 2020. Effects of climate change on pup growth and survival in a cooperative mammal, the meerkat. *Functional Ecology*, 34(1), pp.194-202) from thermal effects on lactating females.
16. Line 90: reduced time for foraging is a problem only if it requires foraging all day to meet the daily energy requirement. Does it require foraging all day, for banded mongoose in equatorial habitats?
17. Line 94: you would make the same prediction if high ambient heat load compromised the physiology of lactation.
18. Line 97-99: the implication is that food density was inadequate when rainfall was low, even in your equatorial habitat. What independent evidence do you have that the lactating females couldn't meet their energy needs fully even when rainfall was low?
19. Line 113: were there any heatwaves in your study period?
20. Line 115: more abundant at higher rainfall doesn't necessarily mean inadequate at lower rainfall.
21. Line 117: what other variables did the weather station measure? You have not reported any. Did you have access to any measure of radiation, water vapour pressure or wind speed? Convective heat loss could have been higher at the higher air temperatures than at the lower ones if wind speed increased sufficiently at times when air temperature was high. So, depending on the wind, lactating females could have dissipated heat to the air more easily at the higher air temperatures.
22. Line 120-122: describe your "methods" in the past tense. You are telling us what you have done, not what you are doing.
23. Line 128: when the pups were caught by hand, you had an ideal opportunity to measure their rectal temperatures, which would have allowed you to test pup body temperature as a covariate, and to eliminate any febrile pups, which was important because fever induces anorexia.
24. Line 132: if the balance accuracy was 1.5 g, how could you measure mass to a tenth of a gram and its SE to a hundredth of a gram? Also, SD not SE is the appropriate statistic here, because you are concerned with variance within the population not between populations.
25. Line 146: what was the accuracy of the thermometer in the weather station? Giving air temperatures to the second decimal place implies that the thermometers were accurate to 1/100°C. In the supplemental information, you gave some temperatures to eight decimal places, when your thermometer probably wasn't accurate even to one decimal place.
26. Line 146: again, giving rainfall to two decimal places implies that the rain gauges were accurate to 1/100 mm.

27. Line 146-149: I think that the appropriate covariate would have been ratio of number of pups to number of lactating females.

28. Line 175, Figure 1: were the slopes of the regression lines significantly different to zero? If they were, why would pup mass decrease with increasing prior rainfall at the highest temperature? According to your postulates, that trend implies less maternal food intake at the higher rainfall. Why? Or is there another factor at play?

29. Line 175, Figure 1: though I think that air temperature was relatively unimportant both in determining heat load (radiation would have much more important) on the lactating females, and in determining their capacity to dissipate body heat (water vapour pressure would have been much more important), if you want to make an argument based on air temperature, you need to show that the sets of temperatures represented by your three means were different statistically.

30. Line 176: again, implying that you could measure air temperature to an accuracy of 0.01°C.

31. Line 181: only if the thermoregulation of the lactating animals was compromised by the higher air temperature. We don't know what their surface temperature was (it can be measured remotely by thermography) but it would have been high when they were in the sun, perhaps even higher than deep-body temperature. Let's say 40°C. Then the gradient for convective heat loss was $40 - 28.3 = 11.7^\circ\text{C}$ at your lowest temperature and $40 - 30.9 = 9.1^\circ\text{C}$ at your highest temperature. So convective heat loss would have differed by only about 20%, and that assumes that wind speed was the same at all air temperatures. Even if the animals weren't cooling by evaporation, that's a small difference. To be consistent with HDL theory, you would have to say that the lactating females were "forced to suppress exothermic processes" (line 183, i.e. to reduce their lactation sufficiently to prevent the pups growing) because they "knew" that their cooling by convection was going to be reduced by 20%, an amount easily compensable by evaporative cooling.

32. Line 189-190: so, you are saying that air temperatures reaching a maximum of 30.9°C outside the den might have imposed a risk of harmful hyperthermia on pups in the den. In adult slender mongoose in the lab, rectal temperature was elevated but held constant when the air temperature was 38°C permanently (Kamau et al. 1979).

33. Line 192: there would have been benefit only when the pups were in the den. As soon as they were outside, and under radiant heat load, a high surface area to mass ratio would have been a disadvantage. The larger pups would have been better off outside the den.

34. Line 199: Van de Ven et al. 2020 is relevant to this paragraph.

35. Line 209: observing pups in the den may have been beyond your resources, but It certainly was possible, with fibre-optic cameras.

36. Line 212: I don't think that the comparisons discussed in this paragraph are relevant. Your pups did not experience high temperatures in early ex-utero development.

37. Line 222-224: your analysis did not confirm an effect of temperature, but only of a temperature/rainfall interaction.

38. Line 224: I do not understand why would you expect high food abundance to compensate for high ambient heat load? High food abundance presumable would lead to more milk consumption and higher metabolic heat production. If you are working within HDL theory, how does that compensate for high ambient heat load?

Review form: Reviewer 3

Recommendation

Major revision is needed (please make suggestions in comments)

Scientific importance: Is the manuscript an original and important contribution to its field?

Good

General interest: Is the paper of sufficient general interest?

Good

Quality of the paper: Is the overall quality of the paper suitable?

Good

Is the length of the paper justified?

Yes

Should the paper be seen by a specialist statistical reviewer?

No

Do you have any concerns about statistical analyses in this paper? If so, please specify them explicitly in your report.

No

It is a condition of publication that authors make their supporting data, code and materials available - either as supplementary material or hosted in an external repository. Please rate, if applicable, the supporting data on the following criteria.

Is it accessible?

Yes

Is it clear?

Yes

Is it adequate?

Yes

Do you have any ethical concerns with this paper?

No

Comments to the Author

I have read the manuscript submitted to Biology Letters by Monil Khera and colleagues with great interest and enthusiasm. The manuscript is sound and timely as it explores an exciting question of the heat dissipation limits (HDL) during lactation in the context of rising ambient temperatures under a climate change scenario, focusing on wild population of mammals (banded mongooses) living in non-seasonal environment near the Equator. The Authors conclude that the higher ambient temperatures (T_a) act to restrict the heat dissipation capacity of lactating females, which presumably reduces their milk production, leading to smaller pups being weaned. As such, the data would support the HDL hypothesis. However, there are several questions to answer and clarify before the Authors publish their work.

1) Both the Title and the Abstract refer to the smaller pups at higher T_a , but the Reader needs to somehow get it from the interaction plot between pup mass and rainfall for 3 different T_a categories (Fig. 1). That may be confusing, as the pups at highest T_a but at low rainfall levels (0-2 mm) are actually heavier than pups at lower T_a . Is this effect significant?

- 2) The interaction plot (Fig. 1) shows the predicted pup mass, based on the LMM model. Would it be possible to visualize and compare the real pup mass data as well, when corrected by age and rainfall?
- 3) The Authors argue that there is a physiological limit acting on the lactating females because rainfall (associated with increased food availability) does not improve pup mass. If there is a cap, and the females reach their maximum milk production, why the pups are getting smaller with increasing rainfall? Once the females reach their maximum milk production, the pup mass should be independent from rainfall and stay 'flat', unless some other mechanisms are involved.
- 4) It remains unclear why the modelling of pup body mass was based on Ta data from lactation while the rainfall data from pregnancy. Are banded mongooses capital or income breeders? If they are income breeders, then they forage extensively during lactation, and perhaps the rainfall data should cover lactation not pregnancy. If they are capital breeders, perhaps both Ta and rainfall data should cover pregnancy as this is the time they would forage extensively to build up body reserves prior to lactation.
- 5) Do the females move between groups? The Supplementary data show 3 females contributing to group 1B and 1N, although at different years.
- 6) It is unclear why the Authors think that pups may be affected by high Ta to similar extent as females while in the den (189-192). Pups have much higher surface-to-mass ratios than females and females will overheat much faster than young. As a result, the pup's optimum Ta would be higher than the mother's optimum Ta.
- 7) If the Authors think that lactating females are limited by Ta outside the den (while foraging), would it be possible to show the differences in foraging time between lower and higher Ta?

Decision letter (RSBL-2023-0328.R0)

11-Aug-2023

Dear Dr Nichols,

The Handling Editor assigned to your paper ("Small increases in temperature reduce offspring mass in an equatorial mammal") has now received comments from reviewers. We would like you to revise your paper in accordance with the referee and Handling Editor suggestions which can be found below (not including confidential reports to the Editor). Please note this decision does not guarantee eventual acceptance. Please endeavour to fully incorporate any revisions the referees, Editor and editorial office have suggested while keeping the paper within journal limits. If you have any questions please do get in touch, as any suggested revisions that are not made may result in a delay to your manuscript.

Please note that Biology Letters normally only allows one round of revisions to take place, so please do take care to incorporate all the suggestions made, or clear reasons as to why you have chosen not to in your 'author response to the referees' document. More revisions may be allowed at the discretion of the Editorial Board.

Please submit a copy of your revised paper within 14 days - if we do not hear from you within this time then it will be assumed that the paper has been withdrawn. In exceptional circumstances, extensions may be possible if agreed with the Editorial Office in advance. Once

submitted your paper may be returned to the previous referees, or new ones if these are unavailable.

To revise your manuscript, log into <http://mc.manuscriptcentral.com/bl> and enter your Author Centre, where you will find your manuscript title listed under "Manuscripts with Decisions." Under "Actions," click on "Create a Revision." Your manuscript number has been appended to denote a revision. Revise your manuscript and upload a new version through your Author Centre.

When submitting your revised manuscript, you must respond to the comments made by the referees and upload a file "Response to Referees" in "Section 2 - File Upload". Please use this to document how you have responded to the comments, and the adjustments you have made. In order to expedite the processing of the revised manuscript, please be as specific as possible in your response.

Information on language editing services can be found on our website - <https://royalsociety.org/journals/authors/language-polishing/>.

Once again, thank you for submitting your manuscript to Biology Letters and I look forward to receiving your revision. If you have any questions at all, please do not hesitate to get in touch.

Yours sincerely,
Surayya Johar
Publishing Editor, Biology Letters
biologyletters@royalsociety.org

Handling Editor Comments to Author:

All three reviewers feel you have a fascinating model system on which you are studying, but the reviewers showed varying levels of support for the manuscript. Reviewer 1 is extremely positive and supports publication. Reviewer 2 is very critical of the study and has requested a significant revision. In essence the reviewer feels that the paper is poorly conceived and, the data are not appropriately analysed. While reviewer 3 has requested significant revisions. As a consequence I feel the paper requires a significant revision. Please would you kindly revise your paper taking cognizance of all points raised in the review process. I would appreciate a detailed covering letter highlighting how you have taken on board the points raised. I think the requests are straightforward and will not reiterate them in this decision letter. I look forward to receiving a revised version of this interesting paper.

Reviewers' Comments to Author:

Referee: 1

Comments to the Author(s)

This manuscript, "Small increases in temperature reduce offspring mass in an equatorial mammal" is exceptionally well thought out, designed, written and reasoned. I have only a two minor edits and one suggestion.

1) on line 162: "at after" the "at" could be deleted, since it is redundant.

2) line 189: "pups" should be "pups'" since this is a possessive word as written.

My suggestion is that the concept of a heat dissipation limit appeared a number of years before papers by Speakman and Krol. The authors may want to check out a paper by Porter et al. (Porter et al. 1994). Figures 15, 22 and 23 in that paper casts thermodynamic constraints on energetics due to climate in a much broader context, including the material presented in this manuscript.

Porter, W., J. Munger, W. Stewart, S. Budaraju, and J. Jaeger. 1994. Endotherm Energetics - From a Scalable Individual-Based Model to Ecological Applications. *Australian Journal of Zoology* 42:125-162.

Referee: 2

Comments to the Author(s)

You have a special, and perhaps unique, data set of body masses, at emergence from the den, of 215 banded mongoose pups, all of which had experienced similar, benign, thermal environments in the den, and all of which had had similar access to food ("pups are suckled indiscriminately by multiple lactating females", line 84). Mean mass at weighing was 189 g but body masses at emergence varied widely, from 93g to 307 g (supplemental information) or 113% of mean mass. Age at weighing varied by about 10 days and age was associated very powerfully with mass (Table 1). Assuming that other factors in the den that could have affected mass at emergence (e.g. incidence of disease, predation) were similar for all pups, it is plausible that differences in mass were associated with the amount of milk delivered by the multiple lactating females. It also is plausible that the amount of milk delivered was associated with the weather during lactation, which would have affected both the thermoregulation of the lactating females but also the amount of food available to them, because the abundance of (predominantly invertebrate) prey increased with increasing rainfall.

It is whether pup mass was associated with the weather to which the lactating females were exposed that you set out to analyse. You concluded, using LMM analysis, that, on their own, neither the maximum air temperature over the period of lactation nor the rainfall at a time earlier, to take account of latency in consequences for prey abundance, had any significant effect on pup mass, but the interaction between maximum temperature and earlier rainfall indeed did so.

Unfortunately, in my opinion, you were so committed to finding support for the heat dissipation limit (HDL) theory of Speakman and Król (reference #9) that you failed to address the data before you on its merits and you overlooked many deficiencies in the arguments that you used to reach your conclusions. In my view, I do not believe there was any solid evidence provided for factors contributing to variability in pup mass at emergence other than age. Also in my view, HDL theory warranted no more than a paragraph in the "discussion".

General comments

1. Animals undoubtedly can be exposed to environments in which they have difficulty dissipating body heat. That's simply a consequence of the laws of physics, not a consequence of HDL theory. Those environments typically will have high radiant heat load and high water-vapour pressure, inhibiting evaporative cooling. There is no information about either radiation or water-vapour pressure in your paper. In such environments, exothermic processes may well cause detrimental hyperthermia (not all hyperthermia is detrimental), and that hyperthermia may reduce or eliminate the exothermic process. Runners who get too hot have to stop running. That's a feedback process. What's different about HDL theory is that it proposes a feedforward process. HDL proposes that animals, not yet hyperthermic, receive some kind of warning signal about impending thermal threats and reduce exothermic processes in anticipation of running into trouble. So, to claim that reducing an exothermic process like lactation is consistent with HDL, you have to demonstrate that the reduction is anticipatory, a demonstration for which I don't think that your data are suitable. A study design that could test the theory might be one in which animals were under the same ambient thermal load in two situations, in one where ambient heat loads were in the process of increasing (threat coming) and in another where ambient heat loads were in the process of decreasing (no threat coming). Then, according to HDL theory, animals would reduce exothermic processes in the first situation but not the second. So, I don't think that you should construct your paper around HDL.

2. It has been known for at least 70 years that high ambient heat load compromises the physiology of lactation, potentially resulting in massive reduction of milk production, especially in high-lactating animals. See the attached diagram for cows, from Hafez, E.S.E., 1968.

Adaptation of domestic animals, based on 1950 data.

If your lactating females were under high enough heat load, their milk production may well have been reduced without them undertaking any anticipatory action to reduce exothermic processes. We do not know whether they were under high heat load because the important source of heat would have been solar radiation, which was not reported.

3. Your crucial results are presented in Table 1 and Figure 1. I believe the regression lines of Figure 1, drawn without any data points, conceal some problematic properties of your data set. For example, from the supplemental information, when rain was more than 4 mm, there were only four pups for which lactating females were exposed to maximum air temperature less than 28°C but ten for which lactating females were exposed to more than 32°C. When rain was less than 2 mm, there were 17 pups for which lactating females were exposed to maximum air temperature less than 28°C but none for which lactating females were exposed to more than 32°C. So the pups were not distributed anywhere near evenly across combinations of rainfall and maximum air temperature to which the lactating females were exposed, as valid regression analysis requires, though the figure may give the impression that they were. Further, again from the supplemental information, the mass of pups for which the lactating females were exposed to rain less than 2 mm and maximum air temperature less than 28°C was 155 ± 34 (SD) g; according to your formulation those females would have had the least food intake and been able to dissipate metabolic heat most easily. The mass of pups for which the lactating females were exposed to rain more than 4 mm and maximum air temperature more than 32°C was 149 ± 29 g; according to your formulation those females would have had the highest food intake and been able to dissipate metabolic heat least easily. But a t-test shows that those mean masses were not significantly different. So the two most-extreme combinations of temperature and rainfall produced pups of the same mass, which is hard to reconcile with your conclusion that the temperature/rain interaction was the weather factor with which changes in mass were associated.

4. The single measure of the status of the thermal environment that you used was the maximum air temperature over the period of exposure. But you point out (lines 106-109) that the lactating females were not active in the middle of the day, which means that they were not exposed to those maximum temperatures. The appropriate temperature to enter into your analysis, I believe, would be the temperature during the foraging bouts, say temperature for the three hours after dawn and the three hours before dusk, averaged over all the days of the exposure.

5. Were rainfall and maximum air temperature independent variables? I'd like to see a plot of rainfall against maximum air temperature. Also, one way in which rainfall and ambient heat load on the animals could have interacted would be if cloud cover increased in the rainier months and so reduced solar radiation. Without a measure of radiation, you can't test that possibility directly. However, I believe that you should re-run your LMM with season as a covariate. Since conventional seasons aren't relevant at the equator, I think that you should designate four seasons around rainfall, for example Season 1 = March-May, Season 2 = June-August, Season 3 = September-November, Season 4 = December- February. That re-run also should use average temperature at the time of foraging and not maximum air temperature.

6. You say (lines 79-80), I believe correctly, that "Studying equatorial species can therefore allow us to decouple the impacts of temperature variation and food supply on energy dynamics". However, studying equatorial species makes it very difficult to discover effects resulting from differences in the thermal status of the environment, because that thermal status varies so much less on the equator than it does at higher latitudes. One of the reasons for temperature not turning up as a factor on its own in your LMM might be that there just was not enough variation in temperature.

7. You have ignored evaporative cooling. Evaporative cooling is by far the most-effective avenue for dissipating body heat, and, except when ambient water-vapour pressure is too high, can dissipate the heat produced by strongly exothermic processes, including heavy exercise.

Evaporative cooling certainly can dissipate, easily, the heat produced by lactation, and sustained evaporative cooling is perfectly possible in equatorial environments where water is readily available. I am aware that banded mongoose are reluctant drinkers but there will be plenty of water available in their prey. I do not know of measurements of evaporative cooling in banded mongoose, but it has been described, and is powerful, in slender mongoose, the range of which overlaps with that of banded mongoose (Kamau, J.M., Johansen, K. and Maloiy, G.M.O., 1979. Thermoregulation and standard metabolism of the slender mongoose (*Herpestes sanguineus*). *Physiological Zoology*, 52(4), pp.594-602), as well as in suricates (Müller, E.F. and Lojewski, U., 1986. Thermoregulation in the meerkat (*Suricata suricatta* Schreber, 1776). *Comparative Biochemistry and physiology. A, Comparative Physiology*, 83(2), pp.217-224). If the banded mongoose has similar evaporatively cooling capacity, and provided ambient water vapour pressure was not too high, your lactating females would have no difficulty dissipating body heat, including heat produced during lactation, even at your hottest air temperature, and even under high solar radiation..

Line by line comments

8.Line 28: equatorial species experience similar temperatures year-round, but those living at high altitude on the equator do not experience high temperatures year-round.

9.Line 44: there are many other strategies that fall under the rubric of “behavioural thermoregulation”.

10.Line 45: ceasing foraging in hot periods does not necessarily reduce energy intake. See Hetem, R.S. et al., 2012. Activity re-assignment and microclimate selection of free-living Arabian oryx: responses that could minimise the effects of climate change on homeostasis? *Zoology*, 115(6), pp.411-416.

11.Line 71: again, not at high altitude.

12.Line 75: you need to make clear that, in temperate environments, increasing temperature is associated with increasing food, because in arid environments, increasing temperature is associated with less food.

13.Line 81: against what alternative or alternatives did you test HDL theory? I do not believe that you were testing the theory. You were seeking support for it.

14.Line 86: underground rearing buffers the pups much more against the effects of solar radiation than it buffers then against high air temperature.

15.Line 87: what underground rearing makes possible is separating thermal effects on pups (see van de Ven et al., 2020. Effects of climate change on pup growth and survival in a cooperative mammal, the meerkat. *Functional Ecology*, 34(1), pp.194-202) from thermal effects on lactating females.

16. Line 90: reduced time for foraging is a problem only if it requires foraging all day to meet the daily energy requirement. Does it require foraging all day, for banded mongoose in equatorial habitats?

17. Line 94: you would make the same prediction if high ambient heat load compromised the physiology of lactation.

18.Line 97-99: the implication is that food density was inadequate when rainfall was low, even in your equatorial habitat. What independent evidence do you have that the lactating females couldn't meet their energy needs fully even when rainfall was low?

19.Line 113: were there any heatwaves in your study period?

20.Line 115: more abundant at higher rainfall doesn't necessarily mean inadequate at lower rainfall.

21.Line 117: what other variables did the weather station measure? You have not reported any. Did you have access to any measure of radiation, water vapour pressure or wind speed? Convective heat loss could have been higher at the higher air temperatures than at the lower ones if wind speed increased sufficiently at times when air temperature was high. So, depending on the wind, lactating females could have dissipated heat to the air more easily at the higher air temperatures.

22.Line 120-122: describe your "methods" in the past tense. You are telling us what you have done, not what you are doing.

23.Line 128: when the pups were caught by hand, you had an ideal opportunity to measure their rectal temperatures, which would have allowed you to test pup body temperature as a covariate, and to eliminate any febrile pups, which was important because fever induces anorexia.

24.Line 132: if the balance accuracy was 1.5 g, how could you measure mass to a tenth of a gram and its SE to a hundredth of a gram? Also, SD not SE is the appropriate statistic here, because you are concerned with variance within the population not between populations.

25.Line 146: what was the accuracy of the thermometer in the weather station? Giving air temperatures to the second decimal place implies that the thermometers were accurate to 1/100°C. In the supplemental information, you gave some temperatures to eight decimal places, when your thermometer probably wasn't accurate even to one decimal place.

26.Line 146: again, giving rainfall to two decimal places implies that the rain gauges were accurate to 1/100 mm.

27.Line 146-149: I think that the appropriate covariate would have been ratio of number of pups to number of lactating females.

28.Line 175, Figure 1: were the slopes of the regression lines significantly different to zero? If they were, why would pup mass decrease with increasing prior rainfall at the highest temperature? According to your postulates, that trend implies less maternal food intake at the higher rain fall. Why? Or is there another factor at play?

29.Line 175, Figure 1: though I think that air temperature was relatively unimportant both in determining heat load (radiation would have much more important) on the lactating females, and in determining their capacity to dissipate body heat (water vapour pressure would have been much more important), if you want to make an argument based on air temperature, you need to show that the sets of temperatures represented by your three means were different statistically.

30.Line 176: again, implying that you could measure air temperature to an accuracy of 0.01°C.

31.Line 181: only if the thermoregulation of the lactating animals was compromised by the higher air temperature. We don't know what their surface temperature was (it can be measured remotely by thermography) but it would have been high when they were in the sun, perhaps even higher than deep-body temperature. Let's say 40°C. Then the gradient for convective heat loss was $40 - 28.3 = 11.7^\circ\text{C}$ at your lowest temperature and $40 - 30.9 = 9.1^\circ\text{C}$ at your highest temperature. So

convective heat loss would have differed by only about 20%, and that assumes that wind speed was the same at all air temperatures. Even if the animals weren't cooling by evaporation, that's a small difference. To be consistent with HDL theory, you would have to say that the lactating females were "forced to suppress exothermic processes" (line 183, i.e. to reduce their lactation sufficiently to prevent the pups growing) because they "knew" that their cooling by convection was going to be reduced by 20%, an amount easily compensable by evaporative cooling.

32.Line 189-190: so, you are saying that air temperatures reaching a maximum of 30.9°C outside the den might have imposed a risk of harmful hyperthermia on pups in the den. In adult slender mongoose in the lab, rectal temperature was elevated but held constant when the air temperature was 38°C permanently (Kamau et al. 1979).

33.Line 192: there would have been benefit only when the pups were in the den. As soon as they were outside, and under radiant heat load, a high surface area to mass ratio would have been a disadvantage. The larger pups would have been better off outside the den.

34.Line 199: Van de Ven et al. 2020 is relevant to this paragraph.

35.Line 209: observing pups in the den may have been beyond your resources, but It certainly was possible, with fibre-optic cameras.

36.Line 212: I don't think that the comparisons discussed in this paragraph are relevant. Your pups did not experience high temperatures in early ex-utero development.

37.Line 222-224: your analysis did not confirm an effect of temperature, but only of a temperature/rainfall interaction.

38.Line 224: I do not understand why would you expect high food abundance to compensate for high ambient heat load? High food abundance presumable would lead to more milk consumption and higher metabolic heat production. If you are working within HDL theory, how does that compensate for high ambient heat load?

Referee: 3

Comments to the Author(s)

I have read the manuscript submitted to Biology Letters by Monil Khera and colleagues with great interest and enthusiasm. The manuscript is sound and timely as it explores an exciting question of the heat dissipation limits (HDL) during lactation in the context of rising ambient temperatures under a climate change scenario, focusing on wild population of mammals (banded mongooses) living in non-seasonal environment near the Equator. The Authors conclude that the higher ambient temperatures (T_a) act to restrict the heat dissipation capacity of lactating females, which presumably reduces their milk production, leading to smaller pups being weaned. As such, the data would support the HDL hypothesis. However, there are several questions to answer and clarify before the Authors publish their work.

1) Both the Title and the Abstract refer to the smaller pups at higher T_a , but the Reader needs to somehow get it from the interaction plot between pup mass and rainfall for 3 different T_a categories (Fig. 1). That may be confusing, as the pups at highest T_a but at low rainfall levels (0-2 mm) are actually heavier than pups at lower T_a . Is this effect significant?

2) The interaction plot (Fig. 1) shows the predicted pup mass, based on the LMM model. Would it be possible to visualize and compare the real pup mass data as well, when corrected by age and rainfall?

3) The Authors argue that there is a physiological limit acting on the lactating females because rainfall (associated with increased food availability) does not improve pup mass. If there is a cap,

and the females reach their maximum milk production, why the pups are getting smaller with increasing rainfall? Once the females reach their maximum milk production, the pup mass should be independent from rainfall and stay 'flat', unless some other mechanisms are involved.

4) It remains unclear why the modelling of pup body mass was based on Ta data from lactation while the rainfall data from pregnancy. Are banded mongooses capital or income breeders? If they are income breeders, then they forage extensively during lactation, and perhaps the rainfall data should cover lactation not pregnancy. If they are capital breeders, perhaps both Ta and rainfall data should cover pregnancy as this is the time they would forage extensively to build up body reserves prior to lactation.

5) Do the females move between groups? The Supplementary data show 3 females contributing to group 1B and 1N, although at different years.

6) It is unclear why the Authors think that pups may be affected by high Ta to similar extent as females while in the den (189-192). Pups have much higher surface-to-mass ratios than females and females will overheat much faster than young. As a result, the pup's optimum Ta would be higher than the mother's optimum Ta.

7) If the Authors think that lactating females are limited by Ta outside the den (while foraging), would it be possible to show the differences in foraging time between lower and higher Ta?

Author's Response to Decision Letter for (RSBL-2023-0328.R0)

See Appendix B.

RSBL-2023-0328.R1

Review form: Reviewer 1

Recommendation

Accept as is

Scientific importance: Is the manuscript an original and important contribution to its field?

Good

General interest: Is the paper of sufficient general interest?

Excellent

Quality of the paper: Is the overall quality of the paper suitable?

Excellent

Is the length of the paper justified?

Yes

Should the paper be seen by a specialist statistical reviewer?

No

Do you have any concerns about statistical analyses in this paper? If so, please specify them explicitly in your report.

No

It is a condition of publication that authors make their supporting data, code and materials available - either as supplementary material or hosted in an external repository. Please rate, if applicable, the supporting data on the following criteria.

Is it accessible?

Yes

Is it clear?

Yes

Is it adequate?

Yes

Do you have any ethical concerns with this paper?

No

Comments to the Author

Great job on a challenging problem. I thoroughly enjoyed not only the manuscript but your responses to all of us.

Review form: Reviewer 2

Recommendation

Major revision is needed (please make suggestions in comments)

Scientific importance: Is the manuscript an original and important contribution to its field?

Good

General interest: Is the paper of sufficient general interest?

Good

Quality of the paper: Is the overall quality of the paper suitable?

Marginal

Is the length of the paper justified?

Yes

Should the paper be seen by a specialist statistical reviewer?

No

Do you have any concerns about statistical analyses in this paper? If so, please specify them explicitly in your report.

Yes

It is a condition of publication that authors make their supporting data, code and materials available - either as supplementary material or hosted in an external repository. Please rate, if applicable, the supporting data on the following criteria.

Is it accessible?

Yes

Is it clear?

Yes

Is it adequate?

Yes

Do you have any ethical concerns with this paper?

No

Comments to the Author

See attached file. (See Appendix C)

Review form: Reviewer 3

Recommendation

Accept with minor revision (please list in comments)

Scientific importance: Is the manuscript an original and important contribution to its field?

Excellent

General interest: Is the paper of sufficient general interest?

Good

Quality of the paper: Is the overall quality of the paper suitable?

Good

Is the length of the paper justified?

Yes

Should the paper be seen by a specialist statistical reviewer?

No

Do you have any concerns about statistical analyses in this paper? If so, please specify them explicitly in your report.

No

It is a condition of publication that authors make their supporting data, code and materials available - either as supplementary material or hosted in an external repository. Please rate, if applicable, the supporting data on the following criteria.

Is it accessible?

Yes

Is it clear?

Yes

Is it adequate?

Yes

Do you have any ethical concerns with this paper?

No

Comments to the Author

The Authors have revised the manuscript sufficiently to be considered for publication in Biology Letters. Some minor suggestions and editorial changes are listed below.

- 1) Title: it would be beneficial to add 'ambient' or 'air' temperature, to distinguish it from 'body' or 'den' temperature. Furthermore, 'offspring body mass' may be more accurate, in case 'offspring mass' is understood as 'litter mass', i.e., mass of all pups per den.
- 2) Abstract & Results/Discussion: it would be important to present the Reader with some magnitude of changes in T_a and pup body mass. How small are the changes in T_a to have an impact on pup body mass? How big are the changes in pup body mass?
- 3) Abstract: Unclear what the Authors mean by the 'opportunity costs'? This term is never discussed in the main part of the paper.
- 4) Figure Legend: unclear how it is evident from CIs that there are no effects of rainfall on pup mass at higher T_a .
- 5) Number and unit should be separated by space. Hyphens, en dashes and em dashes mixed.

Decision letter (RSBL-2023-0328.R1)

16-Oct-2023

Dear Dr Nichols

On behalf of the Editor, I am pleased to inform you that your Manuscript RSBL-2023-0328.R1 entitled "Small increases in temperature reduce offspring mass in an equatorial mammal" has been accepted for publication in *Biology Letters* subject to minor revision in accordance with the referee suggestions. Please find the referees' comments below.

The reviewer(s)/Handling Editor have recommended publication, but also suggest some minor revisions to your manuscript. Therefore, I invite you to respond to the comments and revise your manuscript. Please endeavour to fully incorporate any revisions the referees, Editor and editorial office have suggested while keeping the paper within journal limits. If you have any questions please do get in touch, as any suggested revisions that are not made may result in a delay to your manuscript.

Because the schedule for publication is very tight, it is a condition of publication that you submit the revised version of your manuscript within 7 days. If you do not think you will be able to meet this date please let me know immediately. Failure to do so may cause your manuscript to miss its publication slot and cause severe delays in the publication of your manuscript.

To revise your manuscript, log into <https://mc.manuscriptcentral.com/bland> and enter your Author Centre, where you will find your manuscript title listed under "Manuscripts with Decisions." Under "Actions," click on "Create a Revision." You will be unable to make your revisions on the originally submitted version of the manuscript. Instead, revise your manuscript and upload a new version through your Author Centre.

When submitting your revised manuscript, you will be able to respond to the comments made by the referee(s) and upload a file "Response to Referees" in "Section 2 - File Upload". You can use this to document any changes you make to the original manuscript.

To minimise any delay to publication, please visit our Author Information page for guidance on formatting your manuscript - <https://royalsocietypublishing.org/rsbl/for-authors>. Information

on language editing services can be found on our website - <https://royalsociety.org/journals/authors/language-polishing/>.

- 1) A text file of the manuscript (tex, txt, rtf, docx or doc), references, tables (including captions) and figure captions. Do not upload a PDF as your "Main Document"
- 2) A separate electronic file of each figure (EPS or print-quality PDF preferred (either format should be produced directly from original creation package), or original software format)
- 3) Included a 100 word media summary of your paper when requested at submission. Please ensure you have entered correct contact details (email, institution and telephone) in your user account
- 4) Included the raw data to support the claims made in your paper. You can either include your data as electronic supplementary material or upload to a repository and include the relevant doi within your manuscript
- 5) Accurate supplementary materials. All supplementary materials accompanying an accepted article will be treated as in their final form. Note that the Royal Society will not edit or typeset supplementary material and it will be hosted as provided. Please ensure that the supplementary material includes the paper details where possible (authors, article title, journal name).

Once again, thank you for submitting your manuscript to Biology Letters and I look forward to receiving your revision. If you have any questions at all, please do not hesitate to get in touch.

Best wishes
 Surayya Johar
 Biology Letters editorial office
biologyletters@royalsociety.org

Handling Editor's Comments to Authors:

The current contribution received mixed responses from the reviewers.. Please revise your paper taking cognizance of the points brought to your attention in the review process. Reviewer 1 has a number of valid points, please try to address these in the revision where possible. I would appreciate a detailed covering letter highlighting how you have taken on board the points raised in the review process. Please use the MS word track changes option to show the corrections you have made in the manuscript.

Board Member's comments:

This paper is investigating the heat dissipation limit theory (HDL) under field conditions in banded mongoose. Similar to reviewer 2 and 3, I thoroughly enjoyed reading this paper and believe that it will make an important contribution to the field.

However, I also encourage the authors to at least integrate part of the reviewer's feedback, such as considerations on whether their results could have in fact been caused by maternal hypothermia, into the discussion.

As an additional minor comment, I also recommend clarifying the use of "high temperature" in the introduction as this could mean both, 'high body temperature' or 'high air/ ambient temperature'.

Comments to Authors:

Referee: 2

Comments to the Author(s)

See attached file

Referee: 1

Comments to the Author(s)

great job on a challenging problem. I thoroughly enjoyed not only the manuscript but your responses to all of us.

Referee: 3

Comments to the Author(s)

The Authors have revised the manuscript sufficiently to be considered for publication in *Biology Letters*. Some minor suggestions and editorial changes are listed below.

1) Title: it would be beneficial to add 'ambient' or 'air' temperature, to distinguish it from 'body' or 'den' temperature. Furthermore, 'offspring body mass' may be more accurate, in case 'offspring mass' is understood as 'litter mass', i.e., mass of all pups per den.

2) Abstract & Results/Discussion: it would be important to present the Reader with some magnitude of changes in T_a and pup body mass. How small are the changes in T_a to have an impact on pup body mass? How big are the changes in pup body mass?

3) Abstract: Unclear what the Authors mean by the 'opportunity costs'? This term is never discussed in the main part of the paper.

4) Figure Legend: unclear how it is evident from CIs that there no effects of rainfall on pup mass at higher T_a .

5) Number and unit should be separated by space. Hyphens, en dashes and em dashes mixed.

Author's Response to Decision Letter for (RSBL-2023-0328.R1)

See Appendix D.

Decision letter (RSBL-2023-328.R2)

02-Nov-2023

Dear Dr Nichols,

I am pleased to inform you that your manuscript entitled "Small increases in ambient temperature reduce offspring body mass in an equatorial mammal" is now accepted for publication in *Biology Letters*. As a reminder, you have provided the following 'Data accessibility statement' (if applicable). Please remember to make any data sets live prior to publication, and update any links as needed when you receive a proof to check. It is good practice to also add data sets to your reference list.

Statement (if applicable): Data are available in the supplementary information (for the review process) and will be uploaded to Dryad upon receipt of the submission link.

You can expect to receive a proof of your article within approximately 5-10 working days. Please contact the editorial office (biologyletters@royalsociety.org) to let us know if you are likely to be away from e-mail contact during that period. Due to rapid publication and an extremely tight schedule, if comments are not received, your paper may experience a delay in publication. Our Press Office will be in touch with the corresponding author a week in advance of publication with the correct date and details of the press embargo.

Biology Letters operates under a continuous publication model. Your article will be published straight into the next open issue and this will be the final version of the paper. As such, it can be cited immediately by other researchers. As the issue version of your paper will be the only version to be published I would advise you to check your proofs thoroughly as changes cannot be made once the paper is published.

Open access

Biology Letters is a hybrid open access journal. Depending on the article type and institutional affiliation, our partner CCC may contact the corresponding author about your open access choices from the email domain @copyright.com (if you have any queries regarding fees, please contact authorfees@royalsociety.org). You are invited to opt for open access, our "author pays" publishing model. Payment of open access fees will enable your article to be made freely available via the Royal Society website as soon as it is ready for publication. For more information about open access, including price, please visit the Publishing website (<https://royalsociety.org/journals/authors/open-access/>). Please note if you are corresponding author from one of our special deals your fees may be covered by the deal – please check here <https://royalsociety.org/journals/authors/read-and-publish/read-publish-agreements/> Transformative agreements | Royal Society (for both editorial and production emails that go to authors)

Some ways to effectively promote your article can also be found here

<https://royalsociety.org/blog/2020/07/promoting-your-latest-paper-and-tracking-your-results/>.

The Royal Society has partnered with Copyright Clearance Center's RightsLink service to allow authors to pay article processing charges or page charges. After your manuscript has been accepted, you will receive an email from CCC with the subject "Please submit your article processing/open access charge(s)/page charges" inviting you to pay your charges or request an invoice. If you request an invoice, it will be sent to you from CCC.

It is important to be cautious about payment scams. If you receive an email or text message requesting payment and have any concerns, we recommend contacting us through our website, rather than clicking on any links.

The Society will never ask you to make a direct payment.

On behalf of the Editors of Biology Letters, we look forward to your continued contributions to the Journal.

Best wishes

Surayya Johar, Publishing Editor

biologyletters@royalsociety.org

<https://royalsocietypublishing.org/journal/rsbl>

Comment from the Handling Editor

Thank you kindly for taking on board as many points as possible requested by reviewer 1, it is really appreciated. The paper is now accepted for publication.

Appendix A

FIG. 6-2. Breed differences in the effect of environmental temperature on milk yield in cattle in controlled temperature laboratory at relative humidity of 40 to 60%. (Drawn by H. D. Johnson from data by Ragdale *et al.*, 1950. *Mo. Agric. Exp. Sta. Res. Bull.* Nos. 471 & 501.)

Hafez, E.S.E., 1968. Adaptation of domestic animals. *Adaptation of domestic animals*. Philadelphia : Lea & Febiger.

Appendix B

Dear Professor David Beerling and Professor Nigel Bennett,

We wish to thank you and the reviewers for your constructive comments and for the opportunity to resubmit our manuscript. We have addressed all comments and detail our responses to each individual point below. Our changes included making clarifications throughout our manuscript and incorporating additional literature. We hope the paper now meets your requirements for publication and look forward to hearing back from you in due course.

Kind regards,
Dr Hazel Nichols & co-authors

Handling Editor Comments to Author:

All three reviewers feel you have a fascinating model system on which you are studying, but the reviewers showed varying levels of support for the manuscript. Reviewer 1 is extremely positive and supports publication. Reviewer 2 is very critical of the study and has requested a significant revision. In essence the reviewer feels that the paper is poorly conceived and, the data are not appropriately analysed. While reviewer 3 has requested significant revisions. As a consequence I feel the paper requires a significant revision. Please would you kindly revise your paper taking cognizance of all points raised in the review process. I would appreciate a detailed covering letter highlighting how you have taken on board the points raised. I think the requests are straightforward and will not reiterate them in this decision letter. I look forward to receiving a revised version of this interesting paper.

Thank you for your thoughtful comments and guidance. We have addressed all of the points raised by the reviewers below. We believe that the concerns of reviewer 2 centre around a misunderstanding of HDL theory, and we hope that our response brings clarification.

Reviewers' Comments to Author:

Referee: 1

Comments to the Author(s)

This manuscript, "Small increases in temperature reduce offspring mass in an equatorial mammal" is exceptionally well thought out, designed, written and reasoned. I have only a two minor edits and one suggestion.

Thank you for your comments and suggested changes to our paper, we have now implemented all of these changes.

1) on line 162: "at after" the "at" could be deleted, since it is redundant.

Done (L157)

2) line 189: "pups" should be "pups'" since this is a possessive word as written.

Done (L178)

My suggestion is that the concept of a heat dissipation limit appeared a number of years before papers by Speakman and Krol. The authors may want to check out a paper by Porter et al. (Porter et al. 1994). Figures 15, 22 and 23 in that paper casts thermodynamic constraints on energetics due to

climate in a much broader context, including the material presented in this manuscript.

Porter, W., J. Munger, W. Stewart, S. Budaraju, and J. Jaeger. 1994. Endotherm Energetics - From a Scalable Individual-Based Model to Ecological Applications. *Australian Journal of Zoology* 42:125-162.

Thank you for the suggested reference. We have added this into the introduction as we start to introduce the HDL theory (L48).

Referee: 2

Comments to the Author(s)

You have a special, and perhaps unique, data set of body masses, at emergence from the den, of 215 banded mongoose pups, all of which had experienced similar, benign, thermal environments in the den, and all of which had had similar access to food ("pups are suckled indiscriminately by multiple lactating females", line 84). Mean mass at weighing was 189 g but body masses at emergence varied widely, from 93g to 307 g (supplemental information) or 113% of mean mass. Age at weighing varied by about 10 days and age was associated very powerfully with mass (Table 1). Assuming that other factors in the den that could have affected mass at emergence (e.g. incidence of disease, predation) were similar for all pups, it is plausible that differences in mass were associated with the amount of milk delivered by the multiple lactating females. It also is plausible that the amount of milk delivered was associated with the weather during lactation, which would have affected both the thermoregulation of the lactating females but also the amount of food available to them, because the abundance of (predominantly invertebrate) prey increased with increasing rainfall.

Thank you for the positive assessment of our dataset.

It is whether pup mass was associated with the weather to which the lactating females were exposed that you set out to analyse. You concluded, using LMM analysis, that, on their own, neither the maximum air temperature over the period of lactation nor the rainfall at a time earlier, to take account of latency in consequences for prey abundance, had any significant effect on pup mass, but the interaction between maximum temperature and earlier rainfall indeed did so.

Note that we do not state that the environmental variables that we measured have no significant impact on pup mass alone. Our prediction based on HDL theory was that we would find an interaction between temperature and rainfall on pup mass, and this interaction is what we tested for. As both variables are involved in a significant interaction, we do not report significance values for their individual effects on pup mass as the parameters for these main effects are unintuitive and often biologically misleading in such cases. Instead, we report the interaction in line with standard practice and recommendations for GLMMs and any other statistical models that includes interactions (e.g. Sokal, R.R. and Rohlf, F.J. 1981. *Biometry*, 2nd edition. W.H. Freeman and Company, San Francisco; Thomas, R., Vaughan, I, and Lello, J. 2013. *Data Analysis with R Statistical Software: A Guidebook for Scientists*. Eco-explore, Cardiff).

Unfortunately, in my opinion, you were so committed to finding support for the heat dissipation limit (HDL) theory of Speakman and Król (reference #9) that you failed to address the data before you on its merits and you overlooked many deficiencies in the arguments that you used to reach your conclusions. In my view, I do not believe there was any solid evidence provided for factors contributing to variability in pup mass at emergence other than age. Also in my view, HDL theory warranted no more than a paragraph in the "discussion".

We had no prior commitments to finding support for the HDL theory. We present an alternative possibility in the introduction (that pup mass is related to the energy available to lactating females via resource availability) that we also test. In fact, our introduction begins with describing this alternative to HDL. Our statistical model demonstrates that temperature and rainfall are associated with pup mass (not only that pup mass increases with age). Given that we set out to test a key prediction from HDL theory and that we find support for it, we feel that it warrants more than a mention in our discussion.

General comments

1. Animals undoubtedly can be exposed to environments in which they have difficulty dissipating body heat. That's simply a consequence of the laws of physics, not a consequence of HDL theory. Those environments typically will have high radiant heat load and high water-vapour pressure, inhibiting evaporative cooling. There is no information about either radiation or water-vapour pressure in your paper. In such environments, exothermic processes may well cause detrimental hyperthermia (not all hyperthermia is detrimental), and that hyperthermia may reduce or eliminate the exothermic process. Runners who get too hot have to stop running. That's a feedback process. What's different about HDL theory is that it proposes a feedforward process. HDL proposes that animals, not yet hyperthermic, receive some kind of warning signal about impending thermal threats and reduce exothermic processes in anticipation of running into trouble. So, to claim that reducing an exothermic process like lactation is consistent with HDL, you have to demonstrate that the reduction is anticipatory, a demonstration for which I don't think that your data are suitable. A study design that could test the theory might be one in which animals were under the same ambient thermal load in two situations, in one where ambient heat loads were in the process of increasing (threat coming) and in another where ambient heat loads were in the process of decreasing (no threat coming). Then, according to HDL theory, animals would reduce exothermic processes in the first situation but not the second. So, I don't think that you should construct your paper around HDL.

Both our understanding (which reviewers 1 and 3 appear to concur with) and that of the published literature of the topic (including from the originators of the hypothesis) is that the HDL theory does not propose that animals 'receive some kind of warning signal' about impending thermal threats. It instead proposes that the ability to perform exothermic processes (such as milk production) is limited by their ability to dissipate body heat because of the temperatures they are experiencing at the time. We test for predictions resulting from this hypothesis in our study. Below, we include a selection of quotes (in green) that are representative of how HDL theory is described in the literature. We hope that this illustrates that many of reviewer 2's comments result from a misinterpretation of the HDL theory, and that addressing this misinterpretation allays the majority of their concerns about the suitability of our dataset to study predictions resulting from HDL theory.

In the first formal description of the HDL theory, Speakman and Król (2010, Maximal heat dissipation capacity and hyperthermia risk: neglected key factors in the ecology of endotherms. *Journal of Animal Ecology*, 79(4), pp.726-746) state that "For endothermic animals, we provide evidence suggesting that an upper boundary on total energy expenditure is imposed by the maximal capacity to dissipate body heat and therefore avoid the detrimental consequences of hyperthermia – the heat dissipation limit (HDL) theory" There is no mention of warning signals or anticipatory action; rather it is the current capacity to dissipate heat is important.

Similarly, other studies of HDL such as Nord and Nilsson (2019. Heat dissipation rate constrains reproductive investment in a wild bird. *Functional Ecology*, 33(2), pp.250-259) have also stated that processes such as lactation are limited by the rate at which heat can be dissipated: for example "This eventually led to the formation of the "heat dissipation limit" (HDL) theory, which suggests that

workload during lactation is limited by the rate at which animals can dissipate excess metabolic heat, regardless of the mode of heat transfer, produced during strenuous work (i.e., increased food processing and milk production) (Król & Speakman, 2003b). Such negative impacts of heat stress on reproductive performance have long been known in the animal production industry (Cam & Kuran, 2004; Quiniou),” and van der Vinne, et al. (2014. Temporal niche switching and reduced nest attendance in response to heat dissipation limits in lactating common voles (*Microtus arvalis*). *Physiology & behavior*, 128, pp.295-302) “The hypothesis derived from these experiments and also evidence from larger mammals is that sustained energy intake is limited by the ability of the organism to dissipate the heat that is being generated when food is metabolized, the heat dissipation limitation theory”

Furthermore, previous experimental tests of HDL theory have manipulated the current ability of animals to dissipate heat (by manipulating current ambient temperatures and/or manipulating fur cover). Such research has not interpreted HDL as requiring a warning or that animals anticipate future conditions. For example “According to the heat dissipation limit (HDL) theory, the SusMR at peak lactation is constrained by the maternal capacity to dissipate body heat. To test that theory, we shaved lactating bank voles (*Myodes glareolus*) to experimentally elevate their capacity for heat dissipation.” (Sadowska, E.T., Król, E., Chrzascik, K.M., Rudolf, A.M., Speakman, J.R. and Koteja, P., 2016. Limits to sustained energy intake. XXIII. Does heat dissipation capacity limit the energy budget of lactating bank voles?. *Journal of Experimental Biology*, 219(6), pp.805-815.)

Accordingly, experimental animals were kept under constant temperatures, rather than slowly increasing or decreasing temperatures “Directly contradicting this prediction, it was shown that during cold exposure (8°C), mice not only elevated food intake but also produced more milk and had heavier offspring than they did at room temperature (21°C) (Johnson & Speakman 2001). Similarly, when mice were exposed to thermoneutral conditions (30°C), they ate less food, produced less milk and had smaller offspring (Król, Johnson & Speakman 2003; Król and Speakman 2003a; Król and Speakman 2003b).” (Speakman, J.R. and Król, E., 2010. Maximal heat dissipation capacity and hyperthermia risk: neglected key factors in the ecology of endotherms. *Journal of Animal Ecology*, 79(4), pp.726-746.)

And results have been interpreted in relation to current ambient temperatures, rather than any anticipated future changes in temperature. For example the following studies transfer animals between two different ambient temperatures, rather than gradually changing the temperature which might allow future changes to be anticipated. “We suggest that this central limitation is the maximal capacity of the animal to dissipate body heat, generated as a by-product of processing food and producing milk. This hypothesis predicts that lactating mice kept at 21°C would not elevate their food intake, whatever the additional demands placed on them (Perrigo, 1987; Johnson et al., 2001c), because ingesting additional food would have made them dangerously hyperthermic. However, when females were transferred to the cold (Johnson and Speakman, 2001), the increased driving gradient between body temperature and ambient temperature relaxed the heat dissipation constraint, and the animals were able to elevate their food intake and use that energy for greater milk production. The critical difference between this viewpoint and previous interpretations is that the heat dissipation limit hypothesis views cold as a factor allowing the animals to overcome a constraint on food intake, while previous interpretations have considered exposure to the cold as an additional burden.” (Król, E. and Speakman, J.R., 2003. Limits to sustained energy intake VII. Milk energy output in laboratory mice at thermoneutrality. *Journal of Experimental Biology*, 206(23), pp.4267-4281.) and “The heat dissipation limitation (HDL) theory suggests that the limits to SusEI at peak lactation might be imposed by the capacity of the animal to dissipate body heat generated as a by-product of processing food and producing milk (Król et al., 2003; Król and Speakman, 2003a; Król and Speakman, 2003b). This idea uniquely explains the simultaneous increase in milk production and

the reproduction performance in the cold.” (Yang, D.B., Li, L., Wang, L.P., Chi, Q.S., Hambly, C., Wang, D.H. and Speakman, J.R., 2013. Limits to sustained energy intake. XIX. A test of the heat dissipation limitation hypothesis in Mongolian gerbils (*Meriones unguiculatus*). *Journal of Experimental Biology*, 216(17), pp.3358-3368.)

2. It has been known for at least 70 years that high ambient heat load compromises the physiology of lactation, potentially resulting in massive reduction of milk production, especially in high-lactating animals. See the attached diagram for cows, from Hafez, E.S.E., 1968. *Adaptation of domestic animals, based on 1950 data.*

If your lactating females were under high enough heat load, their milk production may well have been reduced without them undertaking any anticipatory action to reduce exothermic processes. We do not know whether they were under high heat load because the important source of heat would have been solar radiation, which was not reported.

As we explain above, HDL theory proposes that high temperatures limit the ability of females to lactate. It does not imply anticipatory action.

Unfortunately, we do not have measurements of solar radiation at our field site. However high levels of solar radiation increases air temperature (this is one reason why sunny days are usually hotter than cloudy days, even in the shade), so changes in solar radiation are to a high degree correlated with the temperature data that we have measured. Furthermore, banded mongooses often forage and rest in the shade (e.g. under bushes and trees). Therefore, it cannot be assumed that banded mongooses are directly exposed to solar radiation, even if we had measured it.

As our study was conducted in the wild, where many variables are not controllable or (feasibly) measurable, we acknowledge that we were not able to measure every variable that may have influenced banded mongoose body temperature. Nevertheless, we believe that mean maximum temperatures are likely to represent the temperatures that banded mongooses are exposed to, and are likely to have a strong impact on the ability of lactating females to dissipate heat.

3. Your crucial results are presented in Table 1 and Figure 1. I believe the regression lines of Figure 1, drawn without any data points, conceal some problematic properties of your data set. For example, from the supplemental information, when rain was more than 4 mm, there were only four pups for which lactating females were exposed to maximum air temperature less than 28°C but ten for which lactating females were exposed to more than 32°C. When rain was less than 2 mm, there were 17 pups for which lactating females were exposed to maximum air temperature less than 28°C but none for which lactating females were exposed to more than 32°C. So the pups were not distributed anywhere near evenly across combinations of rainfall and maximum air temperature to which the lactating females were exposed, as valid regression analysis requires, though the figure may give the impression that they were. Further, again from the supplemental information, the mass of pups for which the lactating females were exposed to rain less than 2 mm and maximum air temperature less than 28°C was 155 ± 34 (SD) g; according to your formulation those females would have had the least food intake and been able to dissipate metabolic heat most easily. The mass of pups for which the lactating females were exposed to rain more than 4 mm and maximum air temperature more than 32°C was 149 ± 29 g; according to your formulation those females would have had the highest food intake and been able to dissipate metabolic heat least easily. But a t-test shows that those mean masses were not significantly different. So the two most-extreme combinations of temperature and rainfall produced pups of the same mass, which is hard to reconcile with your conclusion that the temperature/rain interaction was the weather factor with which changes in

mass were associated.

The reviewer is correct in pointing out that pup masses were not evenly distributed across rainfall and temperature. However, this is the case with the vast majority of biological data obtained from a wild system (which is often normally rather than uniformly distributed), and nor is it a requirement of a linear regression; regressions make no assumptions of the distribution of explanatory variables (e.g. Sokal, R.R. and Rohlf, F.J. 1981. *Biometry*, 2nd edition. W.H. Freeman and Company, San Francisco; Thomas, R., Vaughan, I, and Lello, J. 2013. *Data Analysis with R Statistical Software: A Guidebook for Scientists*. Eco-explore, Cardiff).

A t-test on a small selection of our dataset may find no difference between raw pup weights. However, t-tests are not appropriate to analyse our dataset, which is why we choose a more sophisticated analysis that takes into account many variables that are not addressed by a t-test. For example, our analyses account for the age of the pup at weighing (which as the reviewer previously mentioned is important in determining pup mass), along with the sex of the pup, the identity of the mother and the identity of the social group. By simultaneously taking into account other variables that impact on pup weight, and by including continuous variables (rather than artificially splitting data into arbitrary and extreme categories using small subsets of data and then conducting t-tests), we are able to reveal the impact of temperature and rainfall on pup weight. Such regression analyses have been used to analyse ecological datasets for decades and their advantages are described in detail in standard ecological textbooks (Huck, S.W. 2009. *Statistical Misconceptions*. Taylor & Francis. New York, U.S.A; Maindonald, J. and Braun, W.J. 2007. *Data Analysis and Graphics Using R: An Example-Based Approach*, 2nd edition. Cambridge University Press. Cambridge, U.K.; Zar, J.H. 1984. *Biostatistical Analysis*, 2nd edition. Prentice-Hall. New Jersey, U.S.A.).

Furthermore, the t-test comparison the reviewer has conducted also seems to demonstrate a misunderstanding of our predictions and results. The idea that the combination of high temperature and high rainfall is the opposite extreme in comparison to low temperature and rainfall is incorrect. We expect that high temperatures have a negative impact on pup weight, whilst high rainfall has a positive impact (through increased food availability). On the other hand, we expect that low temperature has a positive effect by allowing a greater capacity for heat dissipation whereas a low level of rainfall reduces food availability which would lead to lower pup mass. It is therefore not surprising that a t-test comparing these conditions does not find a significant difference.

4. The single measure of the status of the thermal environment that you used was the maximum air temperature over the period of exposure. But you point out (lines 106-109) that the lactating females were not active in the middle of the day, which means that they were not exposed to those maximum temperatures. The appropriate temperature to enter into your analysis, I believe, would be the temperature during the foraging bouts, say temperature for the three hours after dawn and the three hours before dusk, averaged over all the days of the exposure.

Unfortunately, we do not have long-term data available detailing the temperature during foraging bouts, only daily maximum temperature. However, as the reviewer points out in their other comments, the temperature data we present is the temperature in the shade. Therefore, while banded mongooses often rest in the shade in the middle of the day, the temperature data that we analyse is representative of the temperatures that lactating banded mongooses are exposed to.

5. Were rainfall and maximum air temperature independent variables? I'd like to see a plot of rainfall against maximum air temperature. Also, one way in which rainfall and ambient heat load on the animals could have interacted would be if cloud cover increased in the rainier months and so reduced solar radiation. Without a measure of radiation, you can't test that possibility directly.

However, I believe that you should re-run your LMM with season as a covariate. Since conventional seasons aren't relevant at the equator, I think that you should designate four seasons around rainfall, for example Season 1 = March-May, Season 2 = June-August, Season 3 = September-November, Season 4 = December- February. That re-run also should use average temperature at the time of foraging and not maximum air temperature.

Rainfall and air temperature are not strongly correlated (correlation coefficient 0.081, see the plot below). We have added this information to the manuscript (L147-148). However, given that the seasons that the reviewer suggests above are defined based solely on rainfall, season is by definition dependent on rainfall. It would therefore be analytically inappropriate and introduce extreme covariation of explanatory variables if we were to add 'season' into our model as suggested.

6. You say (lines 79-80), I believe correctly, that “Studying equatorial species can therefore allow us to decouple the impacts of temperature variation and food supply on energy dynamics”. However, studying equatorial species makes it very difficult to discover effects resulting from differences in the thermal status of the environment, because that thermal status varies so much less on the equator than it does at higher latitudes. One of the reasons for temperature not turning up as a factor on its own in your LMM might be that there just was not enough variation in temperature.

Since equatorial species are presumably adapted to smaller variations in temperature, they may be susceptible to temperatures outside quite narrow thermal ranges. Because of this, even small changes in temperature could have a disproportionately strong impact compared to temperature species, which is both amongst our justifications for the study (see lines 72-75) and our titular result. Furthermore, when the significant interaction between temperature and rainfall is removed from our model, we still find a significant negative effect of high temperature on pup weight (estimate = -4.63, std error = 1.89, d.f. = 175.46, t-value = -2.46, p=0.015). However, we do not present this in our manuscript because the effect of temperature on pup weight is better explained by the interaction; the impact of temperature changes substantially depending on rainfall.

7. You have ignored evaporative cooling. Evaporative cooling is by far the most-effective avenue for dissipating body heat, and, except when ambient water-vapour pressure is too high, can dissipate the heat produced by strongly exothermic processes, including heavy exercise. Evaporative cooling certainly can dissipate, easily, the heat produced by lactation, and sustained evaporative cooling is perfectly possible in equatorial environments where water is readily available. I am aware that

banded mongoose are reluctant drinkers but there will be plenty of water available in their prey. I do not know of measurements of evaporative cooling in banded mongoose, but it has been described, and is powerful, in slender mongoose, the range of which overlaps with that of banded mongoose (Kamau, J.M., Johansen, K. and Maloiy, G.M.O., 1979. Thermoregulation and standard metabolism of the slender mongoose (*Herpestes sanguineus*). *Physiological Zoology*, 52(4), pp.594-602), as well as in suricates (Müller, E.F. and Lojewski, U., 1986. Thermoregulation in the meerkat (*Suricata suricatta* Schreber, 1776). *Comparative Biochemistry and physiology. A, Comparative Physiology*, 83(2), pp.217-224). If the banded mongoose has similar evaporatively cooling capacity, and provided ambient water vapour pressure was not too high, your lactating females would have no difficulty dissipating body heat, including heat produced during lactation, even at your hottest air temperature, and even under high solar radiation.

It seems to be a strong statement to say "Evaporative cooling *certainly can* dissipate, easily, the heat produced by lactation" (emphasis ours) and "your lactating females would have no difficulty dissipating body heat" when there appears to be no evidence of this assertion in banded mongooses, nor were any reproductive aspects, such as lactation, included in either of the two studies on other mongoose species that the reviewer points to. These very absolute statements hence appear to be unfounded or at best speculative. There is a further key assumption in this comment that "sustained evaporative cooling is perfectly possible in equatorial environments where water is readily available". While there are some water sources around, access to these will be constrained in various ways (such as territory location) and the dry season is also characterised by less plentiful food such that water from food is unlikely to increase during this time either. We also note that there is little evidence for evaporative cooling as a thermoregulatory strategy in the banded mongooses; we have observed these animals in the wild at a wide range of temperatures (up to 37.5°C) and have not observed panting or other specific exposure of moist mucous membranes or other clear behavioural means to flexibly induce high evaporative water loss.

Moreover, we are hesitant to draw overly strong conclusions about evaporative cooling capacity from the two papers suggested by the reviewer. These deal with small numbers of animals (2-7 slender mongooses and 3-5 meerkats, depending on the particular experiments), and seem to have limited relevance to our study for a range of reasons. For instance, these papers used highly stable lab conditions in captive settings rather than values and variation relevant to wild populations. It is notable that the lab conditions involved prolonged periods and reaching extremely high (and stable) ambient temperatures, 5-6 hours at up to 40°C in meerkats and 10 hours at up to 43°C in slender mongooses, which are unlikely to represent conditions to which these species are exposed to in the wild and are certainly not representative of conditions under which our banded mongoose study population lives. Furthermore, even under these rather artificially extreme experimental conditions, evaporative heat loss does not noticeably increase until ambient temperatures (over an unrealistically long period) reach >35°C, and according to these papers even then it is only barely sufficient to compensate for heat production at ~40°C. These statements ignore the further heat gain from the environment (beyond metabolic heat production, and obviously higher at higher ambient temperatures). Along these lines, the authors of the slender mongooses paper point out that, in fact, "At an ambient temperature of 43°C evaporative heat loss exceeded heat production by about 16%; however, the animal could not increase evaporative heat loss sufficiently to maintain a constant body temperature due to the increased heat gain from the environment", and then again that "[when air temperature is hot, above body temperature] heat loss by evaporation exceeds heat production by about 16%. However, due to heat gain from the environment the deep body temperature continues to rise". The authors of the meerkat paper do not consider heat gain from the environment, but given they state that "The amount of water evaporated at ambient temperature of 40°C was sufficient to dissipate approximately the entire *metabolic heat production*" (emphasis ours) it seems highly likely that the same point holds as for the slender mongooses.

Again, this is without the additional metabolic demands of lactation. Since those authors also concluded that the evaporative heat loss in meerkats was predominantly due to panting which, as we mentioned above, is not apparently present in the banded mongooses, it is yet another reason that these papers do not appear highly informative of our study. In brief, while these papers do provide relevant results for basic thermal physiology in meerkats and slender mongooses (not banded mongooses), they use extreme artificial conditions and small numbers of captive animals that are not lactating, which makes them very difficult to apply with any clarity to the physiological ecology of our study of banded mongooses in the wild.

Moreover, it is difficult to see how our results would have arisen if lactating banded mongooses were in a situation where temperature did not pose a problem to the animals, since we have presented evidence that high temperatures lead to smaller pups even when food and hence energy availability (proxied by rainfall) is high. In short, we find little grounds to believe that evaporative cooling is a sufficiently important component of banded mongoose thermal ecology to pose problems to our results, which (again) also cannot be explained by such a scenario as the reviewer proposes.

Line by line comments

8.Line 28: equatorial species experience similar temperatures year-round, but those living at high altitude on the equator do not experience high temperatures year-round.

This is a good point. We have changed the wording to 'often experience high temperature year-round' (L28) as there is not sufficient space in the abstract to add a more detailed explanation. We have, however, explained this further in the introduction (see our response to comment 11 below).

9.Line 44: there are many other strategies that fall under the rubric of "behavioural thermoregulation".

This is indeed the case. In our introduction, we provide two examples of behavioural thermoregulation; reduced activity and microhabitat selection. These are clearly listed as examples and are not presented as an exhaustive list. As behavioural thermoregulation strategies are not a focus of the manuscript, and due to the concise nature of Biology Letters, we choose to provide two examples here, rather than list all potential methods of behavioural thermoregulation.

10.Line 45: ceasing foraging in hot periods does not necessarily reduce energy intake. See Hetem, R.S. et al., 2012. Activity re-assignment and microclimate selection of free-living Arabian oryx: responses that could minimise the effects of climate change on homeostasis?Zoology, 115(6), pp.411-416.

This is an interesting publication, and we thank the reviewer for directing us toward it. Hetem et al (2012) studied Arabian oryx and found that they shifted from diurnal to nocturnal foraging when temperatures were particularly high (although neither energy intake nor changes in body mass were measured in the study). Banded mongooses do not forage after dark so a shift to nocturnal foraging is unlikely to occur in our study. Nevertheless, we have now clarified that 'animals may cease foraging during hot periods, which can reduce their energy intake' (L46), rather than implying that energy intake is always reduced.

11.Line 71: again, not at high altitude.

We have now clarified on 'Animals living close to the equator at low altitudes experience relatively high temperatures year-round' (L71).

12.Line 75: you need to make clear that, in temperate environments, increasing temperature is associated with increasing food, because in arid environments, increasing temperature is associated with less food.

We have now made this clear 'Furthermore, in temperate regions, high seasonal temperatures are generally associated with an increase in food availability' (L75-76).

13.Line 81: against what alternative or alternatives did you test HDL theory? I do not believe that you were testing the theory. You were seeking support for it.

We test HDL theory against the possibility that energy intake alone influences pup growth (via influencing energy available for the production of milk). As we explain in our predictions 'If HDL limits lactation, we predict that pup growth will decrease under high temperatures independently of resource availability, as females become physiologically constrained in their ability to produce milk. However, if lactation is instead determined by energy intake, we predict that pup growth will increase with increasing rainfall, even when temperatures are high. This is because increasing levels of resource availability should allow lactating females to mitigate against the impact of behavioural thermoregulation by consuming more food in the shorter foraging time available.' (L97-100). We also include the alternative to HDL theory in our opening paragraph, whereby we explain that behavioural thermoregulation in hot conditions can lead to a reduction in energy intake (L43-46).

14.Line 86: underground rearing buffers the pups much more against the effects of solar radiation than it buffers them against high air temperature.

It is correct that being underground provides shade. While we have no data on the temperatures of banded mongoose dens (access is very difficult), studies on other species existing at similar temperature ranges have demonstrated substantial buffering of temperature. For example, below we show Figure 3a from Pike and Mitchell (2013, Burrow-dwelling ecosystem engineers provide thermal refugia throughout the landscape. *Animal Conservation*, 16(6), pp.694-703.), which shows a substantial reduction in peak daytime temperatures just 20-40cm from the entrance of gopher tortoise burrows and almost steady temperatures within just ~60cm from the entrance. We discuss the thermal properties of burrows in our discussion (L184-191).

Figure 3a. Thermal profile of a gopher tortoise burrow at different distances from the burrow entrance on a clear day ... in summer. Ambient temperature was recorded by a datalogger just outside the burrow entrance.

15.Line 87: what underground rearing makes possible is separating thermal effects on pups (see van

de Ven et al., 2020. Effects of climate change on pup growth and survival in a cooperative mammal, the meerkat. *Functional Ecology*, 34(1), pp.194-202) from thermal effects on lactating females.

We believe that this is the point we were making in our manuscript. To increase clarity, we have changed our statement to: 'Underground rearing is likely to buffer pups against direct negative effects of high temperatures, therefore separating thermal effects on pups from those on lactating females. This makes them ideal for studying the indirect effects of high temperatures on pup growth via its effect on milk production.' (L87-90).

16. Line 90: reduced time for foraging is a problem only if it requires foraging all day to meet the daily energy requirement. Does it require foraging all day, for banded mongoose in equatorial habitats?

Banded mongooses in our population generally forage all day, except for the middle of the day (the hottest period) when they rest. We are currently collecting detailed data on changes in foraging in relation to temperature changes. Our preliminary data finds that foraging time and activity levels are reduced in high ambient temperatures (see our response to reviewer 3 below), however we do not have sufficient data on food intake to determine whether lactating females struggle to meet their daily energy needs when their foraging time is reduced (food intake is not easy to determine in a wild system). As we state in our current manuscript, the availability of food in the environment is likely to be of particular importance in determining whether a reduction in foraging time is likely to reduce energy intake. If energy intake by lactating females determines pup weight, we would expect to find that pup weight increases with increasing rainfall (food availability), even when temperatures are high, as females should be able to find sufficient food despite having to reduce their foraging time. However, this is not what we find. Instead, changes in rainfall have little impact on pup weight when temperatures are high, but have the expected positive relationship at lower temperatures. This supports our prediction that heat dissipation rather than food availability limits lactation at high temperatures.

17. Line 94: you would make the same prediction if high ambient heat load compromised the physiology of lactation.

The HDL theory proposes that high ambient temperatures do indeed compromise lactation, so this is not an alternative hypothesis (see our earlier responses).

18. Line 97-99: the implication is that food density was inadequate when rainfall was low, even in your equatorial habitat. What independent evidence do you have that the lactating females couldn't meet their energy needs fully even when rainfall was low?

Directly assessing the impact of rainfall on lactation in our wild study system would require data on calorific intake and the quantity and quality of milk produced, which is unfortunately not feasible to collect from our wild study population (we are neither able to assess the calorific value of the small invertebrates that banded mongooses eat, nor to milk the females). However, we have previously demonstrated that rainfall is positively linked to invertebrate abundance at our study site (Rood J. Population dynamics and food habits of the banded mongoose. *Afr J Ecol.* 1975;13(2):89–111; De Luca D. The socio-ecology of a plural breeding species: The banded mongoose. PhD Thesis, University College London; 1998; Marshall HH, Vitikainen EIK, Mwanguhya F, Businge R, Kyabulima S, Hares M. Lifetime fitness consequences of early-life ecological hardship in a wild mammal population. *Ecol Evol.* 2017;7(6):1712.) and that banded mongoose daily weight gain increases with rainfall in the past 30 days (Marshall HH, Sanderson JL, Mwanguhya F, Businge R, Kyabulima S, Hares M., et al. Variable ecological conditions promote male helping by changing banded mongoose group

composition. *Behav Ecol.* 2016;27(4):978–87.). Furthermore, female weight changes are more susceptible to this short-term variation in rainfall, gaining more weight than males when rainfall is high (Marshall et al., 2016). Together with the results of our current manuscript, our data strongly suggests that rainfall over the prior 30 day period limits the energy intake of female banded mongooses.

19.Line 113: were there any heatwaves in your study period?

We analysed mean daily maximum temperatures (°C) during the lactation period (when pups were aged 0-30 days). Periods of prolonged high temperatures during this period will be reflected in high mean temperatures, however we did not directly quantify heatwaves. To ensure this is clear, we have changed the term 'heatwave' for 'hot conditions' on line 194 and 208 in the discussion. Note that the reference to heatwaves on L107 of the methods (and elsewhere in the introduction and discussion) refer to the terminology used in the studies referenced.

20.Line 115: more abundant at higher rainfall doesn't necessarily mean inadequate at lower rainfall.

See our response to comment 18. Note that here, we are stating a comparative difference in food availability in relation to rainfall that has been observed at our study site.

21.Line 117: what other variables did the weather station measure? You have not reported any. Did you have access to any measure of radiation, water vapour pressure or wind speed? Convective heat loss could have been higher at the higher air temperatures than at the lower ones if wind speed increased sufficiently at times when air temperature was high. So, depending on the wind, lactating females could have dissipated heat to the air more easily at the higher air temperatures.

Unfortunately, we only have access to data on rainfall and air temperature. We did not have access to measures of radiation, water vapour pressure or wind speed. However, our anecdotal field observations are that wind speed does not increase at higher temperatures.

22.Line 120-122: describe your "methods" in the past tense. You are telling us what you have done, not what you are doing.

We have gone over our methods section and have changed to past tense where we describe data collection. However, as our long-term study (and the climate in Uganda) is ongoing, we have retained present tense in some of the sentences describing the study site, climatic conditions, and banded mongoose behaviour, as these are ongoing, and we want to avoid implying that they no longer exist or have changed recently.

23.Line 128: when the pups were caught by hand, you had an ideal opportunity to measure their rectal temperatures, which would have allowed you to test pup body temperature as a covariate, and to eliminate any febrile pups, which was important because fever induces anorexia.

Pups are gently picked up by the scruff of the neck and are placed on the weighing scales (after a very brief look at their genitals for sexing, which are visible as they curl their abdomen in response to being picked up). This process takes 5-10 seconds and mimics the process by which pups are picked up by adults when they carry them. In order to avoid dishabituating the animals and causing undue stress, we do not manipulate the pups during handling, and we do not take any other measurements without anaesthesia. This is in accordance with our ethics permits from the Ugandan Wildlife Authority.

24.Line 132: if the balance accuracy was 1.5 g, how could you measure mass to a tenth of a gram and its SE to a hundredth of a gram? Also, SD not SE is the appropriate statistic here, because you are concerned with variance within the population not between populations.

Pup weights were taken in grams. In this paragraph, we give mean values to 1dp (raw values are to the nearest gram). We have now removed the SE values from this paragraph and have given ranges instead.

25.Line 146: what was the accuracy of the thermometer in the weather station? Giving air temperatures to the second decimal place implies that the thermometers were accurate to 1/100°C. In the supplemental information, you gave some temperatures to eight decimal places, when your thermometer probably wasn't accurate even to one decimal place.

Raw temperature data were measured to 0.1 degree, which we have now explained in the main text (L111). Here, we present mean values rather than raw data, and there is no requirement for summary statistics to be presented at the same precision as the data. Nevertheless, we have now changed all values in the main text to 1dp at your suggestion. In the dataframe, we retain mean values as they are, as reducing the number of decimal places will not qualitatively influence our results.

26.Line 146: again, giving rainfall to two decimal places implies that the rain gauges were accurate to 1/100 mm.

Rainfall was measured to 0.1mm. Here, we present mean values rather than raw data (and see above for comment on the difference between raw data and statistical estimates such as averages). We have now changed them to 1dp and clarified precision (L111)

27.Line 146-149: I think that the appropriate covariate would have been ratio of number of pups to number of lactating females.

We agree that this would be a more appropriate covariate. However, as pups are born underground and remain there for the first ~30 days of their life, we are unable to count the number of pups at birth, and it is likely that some pups die before we are able to observe them. Due to the resulting uncertainty in pup:female ratios at birth, we therefore include in our model the number of lactating females.

28.Line 175, Figure 1: were the slopes of the regression lines significantly different to zero? If they were, why would pup mass decrease with increasing prior rainfall at the highest temperature? According to your postulates, that trend implies less maternal food intake at the higher rainfall. Why? Or is there another factor at play?

The lines on the figure are to illustrate our model's output at different temperatures (for the benefit of readers) as we included temperature in our model as a continuous variable rather than a 3-level factor. The slope of the line at the higher temperature is very slightly negative, but the 95% CIs strongly indicate that the slope is not different from zero at this temperature value. Furthermore, when the 73 pups raised at over Tmax 30°C are modelled separately, there is no significant impact of rainfall on pup weight (GLM; effect = -1.05 SE = 3.82, t = -0.28, P = 0.78), showing that pup weight does not significantly decrease with increasing rainfall for those pups raised at high temperatures. We have now explained in the figure legend that there is no effect of rainfall on pup weight at higher temperatures to clarify this point.

29.Line 175, Figure 1: though I think that air temperature was relatively unimportant both in determining heat load (radiation would have much more important) on the lactating females, and in determining their capacity to dissipate body heat (water vapour pressure would have been much more important), if you want to make an argument based on air temperature, you need to show that the sets of temperatures represented by your three means were different statistically.

The lines on the figure are for illustrative purposes as we included temperature in our model as a continuous variable rather than a 3-level factor. The impact of temperature is tested in the statistical model, rather than in the figure (see the methods and Table 1 for a description of our model). We have added the statement that 'Temperature is categorized here for illustrative purposes and was included as a continuous variable in our analyses but we show the predicted rainfall-pup mass relationship at three temperatures for illustrative purposes' into our figure legend for clarification.

30.Line 176: again, implying that you could measure air temperature to an accuracy of 0.01°C.

Temperatures were measured to 0.1 degree. Here, we present mean values rather than raw data (see comments above). We have now changed them to 1dp.

31.Line 181: only if the thermoregulation of the lactating animals was compromised by the higher air temperature. We don't know what their surface temperature was (it can be measured remotely by thermography) but it would have been high when they were in the sun, perhaps even higher than deep-body temperature. Let's say 40°C. Then the gradient for convective heat loss was $40-28.3 = 11.7^{\circ}\text{C}$ at your lowest temperature and $40-30.9 = 9.1^{\circ}\text{C}$ at your highest temperature. So convective heat loss would have differed by only about 20%, and that assumes that wind speed was the same at all air temperatures. Even if the animals weren't cooling by evaporation, that's a small difference. To be consistent with HDL theory, you would have to say that the lactating females were "forced to suppress exothermic processes" (line 183, i.e. to reduce their lactation sufficiently to prevent the pups growing) because they "knew" that their cooling by convection was going to be reduced by 20%, an amount easily compensable by evaporative cooling.

We do not suggest that banded mongooses 'know' the percentage values that their cooling by convection will be reduced by: as we explain above, the HDL hypothesis is not about predicting future conditions. There seems to be some important assumptions in the comment about what is "easily compensable" (under what conditions, what is the threshold for problems [how hot is too hot], etc.) which appear to be too speculative to address further. Nevertheless, we have changed the phrasing of the opening paragraph of the discussion (L169-177) to emphasise that our findings are consistent with HDL theory (rather than stating that they are a result of HDL).

32.Line 189-190: so, you are saying that air temperatures reaching a maximum of 30.9°C outside the den might have imposed a risk of harmful hyperthermia on pups in the den. In adult slender mongoose in the lab, rectal temperature was elevated but held constant when the air temperature was 38°C permanently (Kamau et al. 1979).

30.9°C was the mean daily maximum temperature +1SD (see Figure 1 legend). The highest mean daily maximum temperature over the 30-day lactation period was 32.1°C. However, this is a mean value rather than an absolute maximum. In this paragraph, we suggest that high temperatures could potentially affect the pups' ability to dissipate heat. We then go on to say that this possibility is unlikely due to the thermal properties of the dens. We have now modified the wording of the start of this paragraph (also following a comment by reviewer 3) to emphasise that we present the

possibility that pups may be directly impacted by high temperatures, but that any direct effects are unlikely.

Note also, as stated above in response to an earlier comment which also addressed the limited direct or quantitative relevance of Kamau et al. (1979) to our study, that those authors did not find that rectal temperature held constant at high temperatures. Specifically, they stated (emphasis ours) "At an ambient temperature of 43°C evaporative heat loss exceeded heat production by about 16%; however, the animal could not increase evaporative heat loss sufficiently to maintain a constant body temperature due to the increased heat gain from the environment", and then again that "[at high temperatures] heat loss by evaporation exceeds heat production by about 16%. However, due to heat gain from the environment the deep body temperature continues to rise". In both cases the authors specifically state that body (rectal) temperatures continued increasing at higher temperatures, in contrast to the reviewer's claim.

33.Line 192: there would have been benefit only when the pups were in the den. As soon as they were outside, and under radiant heat load, a high surface area to mass ratio would have been a disadvantage. The larger pups would have been better off outside the den.

The pups remain in the den for the vast majority of their time for the first 30 days of life. We have now removed our previous comment that pups may benefit from reduced growth under hot conditions, as we agree that it is too speculative.

34.Line 199: Van de Ven et al. 2020 is relevant to this paragraph.

We now include this reference (number 53): 'Similar effects have been found in meerkats, whereby weaned pups have reduced mass gain at high temperatures, but without an apparent reduction in feeding rate (53). However, in our study of banded mongooses, we focused on pups that pups are raised in underground dens pre-weaning' (L181-183).

35.Line 209: observing pups in the den may have been beyond your resources, but It certainly was possible, with fibre-optic cameras.

We thank the reviewer for their suggestion. Over the 20+ years of our study, we have indeed tried to observe pups in the den using such cameras, however due to the positioning of dens (often in inaccessible locations such as dense bushes or underneath buildings), we have been unable to reliably collect such observations. We have changed the phrasing here to emphasise that observations of pups in the den are rarely possible (L200).

36.Line 212: I don't think that the comparisons discussed in this paragraph are relevant. Your pups did not experience high temperatures in early ex-utero development.

Here, we state that high temperatures have been found to have negative effects on the early-life development of other vertebrates. In most studies in the wild, adults and juveniles are both exposed to the similar environmental conditions, which make it difficult to test for indirect impacts on offspring via the effects of high temperatures on adults (as we have done here). However, we believe these studies are sufficiently relevant to warrant mention. We have also added in the Van de Ven, (2020) study on meerkats that the reviewer brought our attention to as we feel this is also relevant here (L204-208).

37.Line 222-224: your analysis did not confirm an effect of temperature, but only of a

temperature/rainfall interaction.

By definition, a variable involved in an interaction has an effect. Therefore, the fact that we found a temperature / rainfall interaction in this model demonstrates an effect of temperature on pup mass, and that the nature of this effect varies depending on the level of rainfall (a proxy for food availability). When rainfall is low, changes in ambient temperature has little / no effect on pup mass however when rainfall is high pup mass is clearly higher at lower ambient temperatures. Note that we also ran the model without the interaction term and found a significant effect of temperature on pup mass (see our response to comment 6), but we do not report it in the manuscript as temperature and rainfall were involved in a significant interaction, which better explains their effect on pup mass than considering them separately.

38.Line 224: I do not understand why would you expect high food abundance to compensate for high ambient heat load? High food abundance presumably would lead to more milk consumption and higher metabolic heat production. If you are working within HDL theory, how does that compensate for high ambient heat load?

The reviewer is correct that HDL theory does not predict that food availability limits lactation at high temperatures, rather that high temperatures reduce lactation regardless of food availability. This is consistent with our results (high temperatures led to low pup weight regardless of rainfall), and is why we suggest that our study supports HDL theory.

In our manuscript, we present (as an alternative to HDL theory) the possibility that high temperatures may lead to reduced pup growth through limiting the ability of lactating females to obtain energy through foraging, for example if they reduce foraging activities due to the need to behaviourally thermoregulate. A reduction in the weight of pups at high temperatures alone therefore does not necessarily provide strong evidence that difficulty dissipating heat limits lactation and leads to reduced pup growth. Rather lactation could be limited by restrictions to energy intake. This is why testing for an interaction between temperature and food availability is key to our study; our finding that high temperatures are associated with reduced pup weight regardless of food availability supports the possibility that heat dissipation, rather than energy intake, causes the reduction in pup weight at high temperatures.

Referee: 3

Comments to the Author(s)

I have read the manuscript submitted to Biology Letters by Monil Khera and colleagues with great interest and enthusiasm. The manuscript is sound and timely as it explores an exciting question of the heat dissipation limits (HDL) during lactation in the context of rising ambient temperatures under a climate change scenario, focusing on wild population of mammals (banded mongooses) living in non-seasonal environment near the Equator. The Authors conclude that the higher ambient temperatures (T_a) act to restrict the heat dissipation capacity of lactating females, which presumably reduces their milk production, leading to smaller pups being weaned. As such, the data would support the HDL hypothesis. However, there are several questions to answer and clarify before the Authors publish their work.

We wish to thank the reviewer for their positive evaluation of our work, including highlighting the timeliness of our study and the importance of exploring the impacts of increasing temperatures on

animal reproduction. We have done our best to answer the queries below and hope that you find our work to be ready for publication.

1) Both the Title and the Abstract refer to the smaller pups at higher Ta, but the Reader needs to somehow get it from the interaction plot between pup mass and rainfall for 3 different Ta categories (Fig. 1). That may be confusing, as the pups at highest Ta but at low rainfall levels (0-2 mm) are actually heavier than pups at lower Ta. Is this effect significant?

The lines on the figure are predictions from our model that represent three points of a changing slope, rather than separate regressions for different temperatures. We plot these as an illustration of the interaction, however we actually fitted temperature as a continuous variable. When attempting to graphically show an interaction between two continuous traits, a common option is the one we have used – choosing a few select values of one variable (in this case temperature at its mean value and one standard deviation either side thereof) and plotting the predicted line for the other variable (rainfall) for each of these. The other common strategy is to use 3D surface plots or similar, but we prefer our approach due to the difficulties of reading 3D plots on 2D surfaces (paper or screens). The three lines on this plot shouldn't be interpreted as distinctly and uniquely meaningful at every position in themselves, but read in combination as a holistic illustration of how the relationship between rainfall and pup mass changes as temperatures increase. We have now clarified our figure legend to better explain our approach.

The slight decrease in pup weight with increasing rainfall for the pups raised in the hottest mean temperatures (the red line on the graph) has CIs that substantially overlap a flat line, strongly suggesting no significant difference. Furthermore, when the 73 pups raised at over Tmax 30°C are modelled separately, there is no significant impact of rainfall on pup weight (GLM; effect = -1.05 SE = 3.82, $t = -0.28$, $P = 0.78$). We have now explained in the figure legend that “Temperature was included as a continuous variable in our analyses but we show the predicted rainfall-pup mass relationship at three temperatures for illustrative purposes. At low and medium temperatures, pup weight increases with rainfall, however at high temperatures, there is no effect of rainfall on pup weight (as evident from the CIs).” We hope that this explanation now avoids any confusion.

2) The interaction plot (Fig. 1) shows the predicted pup mass, based on the LMM model. Would it be possible to visualize and compare the real pup mass data as well, when corrected by age and rainfall?

We have considered adding data points to our figure, but this not straight-forward because (1) raw weight data is not corrected for age at weighing (which has a strong effect) and (2) we collected temperature data on a continuous scale, rather than holding temperature at three different levels, which means that our datapoints don't directly correspond with any of the lines presented on the figure (as explained in our response to comment 1). We could potentially calculate residuals of pup weight for their age to resolve issue 1, but the datapoints would then be on a different scale to the y-axis so would not be comparable to our model output. We have therefore decided to retain the figure as it is, as we believe that it best represents our findings. We have, however, changed how the lines are visualised to make them easier to distinguish when printed in greyscale or for colourblind readers. We have also now included an image of some banded mongoose pups at the appropriate age to illustrate the study system to the reader.

3) The Authors argue that there is a physiological limit acting on the lactating females because rainfall (associated with increased food availability) does not improve pup mass. If there is a cap, and the females reach their maximum milk production, why the pups are getting smaller with increasing

rainfall? Once the females reach their maximum milk production, the pup mass should be independent from rainfall and stay 'flat', unless some other mechanisms are involved.

As we have explained in our response to reviewer 2 and point 1, the lines on our figure are for illustrative purposes (predictions from the models), and both rainfall and temperature were fitted as continuous variables, which we have now clarified in the figure legend. Importantly, there is no detectable effect of rainfall on pup mass at higher temperatures (a key part of our conclusion since more food doesn't translate to bigger pups at high temperatures, unlike at lower temperatures, as predicted by the HDL hypothesis), and we have clarified this in the figure legend.

4) It remains unclear why the modelling of pup body mass was based on Ta data from lactation while the rainfall data from pregnancy. Are banded mongooses capital or income breeders? If they are income breeders, then they forage extensively during lactation, and perhaps the rainfall data should cover lactation not pregnancy. If they are capital breeders, perhaps both Ta and rainfall data should cover pregnancy as this is the time they would forage extensively to build up body reserves prior to lactation.

We do not have a good understanding of to what extent banded mongooses are capital vs income breeders, but our understanding of the system based on ~20 years of research has guided our decisions here. Specifically, rainfall for 30 days prior to birth (rather than rainfall over the lactation period) was used because there is a lag of approximately this duration in the time it takes for rainfall to translate into changes in invertebrate abundance (food availability) at our field site. We used temperature over the lactation period as this is the relevant period for testing the HDL hypothesis – we are looking for direct effects of current temperature here. Perhaps counterintuitively, using rainfall over the 30 days prior to birth (i.e. during pregnancy) actually does relate to the same time period as the temperature data (i.e. during lactation) since rainfall is a time-delayed proxy of food availability. Put another way, we do not predict direct effects of current rainfall on pup weight, but the amount of rain during pregnancy influences the invertebrate abundance later, during the lactation period. We have now made this clear in the methods section (L135-141).

5) Do the females move between groups? The Supplementary Data show 3 females contributing to group 1B and 1N, although at different years.

Yes, while most individuals remain within their natal group, females do occasionally move between groups. Dominant females have been shown to evict subordinate females which then form groups of their own; this is what happened when these three females left group 1B and started group 1N. The territories of 1B and 1N differed, which potentially influenced pup growth, which is why we include social group in the model as a random effect.

6) It is unclear why the Authors think that pups may be affected by high Ta to similar extent as females while in the den (189-192). Pups have much higher surface-to-mass ratios than females and females will overheat much faster than young. As a result, the pup's optimum Ta would be higher than the mother's optimum Ta.

We have now changed the wording of this paragraph to better explain that pups are unlikely to be affected to a similar extent as adults; 'In addition to placing constraints on lactation, high temperatures could also directly affect the pups' ability to dissipate heat causing them to reduce milk intake in order to avoid hyperthermia, although likely to a lesser extent than adults due to the pups' higher surface area to volume ratios' (L178-180). We have also modified this paragraph, following reviewer 2's comments, and hope that our point is now much clearer.

7) If the Authors think that lactating females are limited by T_a outside the den (while foraging), would it be possible to show the differences in foraging time between lower and higher T_a ?

Whilst there is no published study directly showing the effect of high temperature on foraging time in banded mongooses, we are currently in the process of analysing data that we collected in May 2023 to investigate the impact of temperature on foraging behaviour. Our data does indeed show that the proportion of time focal individuals spent foraging reduces during high ambient temperatures. We have included a figure of our preliminary data below, but we are not yet at the point of being able to publish our results. Note that the figure below shows ambient temperature, which varies across the course of a day as well as between days. In the current paper, we analyse mean T_{max} , so we need to be careful not to make quantitative conclusions about foraging behaviour in relation to the current paper based on the below figure. Nevertheless, the basic premise that foraging reduces at higher temperatures is supported by our as-yet unpublished data.

Editorial office comments to authors:

In answer to the editorial office queries:

- 1) We would also like to highlight that the figure is our own, so no permissions are required for its inclusion
- 2) We would like to publish open-access, and or institution (Swansea University) has a read and publish agreement.

Appendix C

For authors, round 2

Thank you for the attention that you have given to my comments. You have resolved several of the problems that I had. The comments that follow relate mostly to those that I regard as unresolved or not fully resolved.

.....

It is whether pup mass was associated with the weather to which the lactating females were exposed that you set out to analyse. You concluded, using LMM analysis, that, on their own, neither the maximum air temperature over the period of lactation nor the rainfall at a time earlier, to take account of latency in consequences for prey abundance, had any significant effect on pup mass, but the interaction between maximum temperature and earlier rainfall indeed did so.

Note that we do not state that the environmental variables that we measured have no significant impact on pup mass alone. Our prediction based on HDL theory was that we would find an interaction between temperature and rainfall on pup mass, and this interaction is what we tested for. As both variables are involved in a significant interaction, we do not report significance values for their individual effects on pup mass as the parameters for these main effects are unintuitive and often biologically misleading in such cases.

A relationship between pup mass and temperature alone is intuitive to me if lactation fails under heat stress. To make your approach clear, I think that you should add a sentence, in an appropriate place, to the effect that “we did not demonstrate, and nor did we seek to test, a relationship between pup mass and either temperature or rainfall as independent variables”.

Unfortunately, in my opinion, you were so committed to finding support for the heat dissipation limit (HDL) theory of Speakman and Król (reference #9) that you failed to address the data before you on its merits and you overlooked many deficiencies in the arguments that you used to reach your conclusions. In my view, I do not believe there was any solid evidence provided for factors contributing to variability in pup mass at emergence other than age. Also in my view, HDL theory warranted no more than a paragraph in the “discussion”.

We had no prior commitments to finding support for the HDL theory. We present an alternative possibility in the introduction (that pup mass is related to the energy available to lactating females via resource availability) that we also test. In fact, our introduction begins with describing this alternative to HDL. Our statistical model demonstrates that temperature and rainfall are associated with pup mass (not only that pup mass increases with age).

That is not the alternative hypothesis that I would have liked to have seen. Given the extensive literature on heat stress and lactation, my alternative hypothesis would have been that lactation is compromised by heat stress leading to maternal hyperthermia, irrespective of food supply. The problem is that you don't have a good measure of heat stress. Your LMM indeed shows that the interaction between temperature and rainfall is associated statistically with pup mass at emergence, but I submit that the effect is weak, as would have been more evident if you have plotted your Fig. 1 over the full range of pup masses (93 -307 g), rather than about 150-250 g.

General comments

1. Animals undoubtedly can be exposed to environments in which they have difficulty dissipating body heat. That's simply a consequence of the laws of physics, not a consequence of HDL theory. Those environments typically will have high radiant heat load and high water-vapour pressure, inhibiting evaporative cooling. There is no information about either radiation or water-vapour pressure in your paper. In such environments, exothermic processes may well cause detrimental hyperthermia (not all hyperthermia is detrimental), and that hyperthermia may reduce or eliminate the exothermic process. Runners who get too hot have to stop running. That's a feedback process. What's different about HDL theory is that it proposes a feedforward process. HDL proposes that animals, not yet hyperthermic, receive some kind of warning signal about impending thermal threats and reduce exothermic processes in anticipation of running into trouble. So, to claim that reducing an exothermic process like lactation is consistent with HDL, you have to demonstrate that the reduction is anticipatory, a demonstration for which I don't think that your data are suitable.

Both our understanding (which reviewers 1 and 3 appear to concur with) and that of the published literature of the topic (including from the originators of the hypothesis) is that the HDL theory does not propose that animals 'receive some kind of warning signal' about impending thermal threats. It instead proposes that the ability to perform exothermic processes (such as milk production) is limited by their ability to dissipate body heat because of the temperatures they are experiencing at the time.

Then what is different between HDL theory and the theory, for which there is plenty of evidence, some of which you cite, that lactation fails under heat stress leading to maternal hyperthermia? I would not regard runners stopping running because they are too hot as an example of HDL in operation.

The key word in the quotation from the original Speakman and Król paper is "avoid": "*upper boundary on total energy expenditure is imposed by the maximal capacity to dissipate body heat and therefore avoid the detrimental consequences of hyperthermia*". How do you "avoid" without pre-emptive action? Similarly, in a later quotation: "*because ingesting additional food would have made them dangerously hyperthermic*". Not ingesting additional food in anticipation of possible hyperthermia is a feedforward process.

The authors cite one of Speakman's elegant shaving experiments. There is another Speakman shaving experiment on voles (Simons, M.J., Reimert, I., van der Vinne, V., Hambly, C., Vaanholt, L.M., Speakman, J.R. and Gerkema, M.P., 2011. Ambient temperature shapes reproductive output during pregnancy and lactation in the common vole (*Microtus arvalis*): a test of the heat dissipation limit theory. *Journal of Experimental Biology*, 214(1), pp.38-49) which concludes "*Shaving fur off dams at 30°C resulted in faster growth of pups; however, no significant increase in food intake and or milk production was detected*".

2. It has been known for at least 70 years that high ambient heat load compromises the physiology of lactation, potentially resulting in massive reduction of milk production, especially in high-lactating animals. See the diagram for cows, from Hafez, E.S.E., 1968. Adaptation of domestic animals, based on 1950 data :

If your lactating females were under high enough heat load, their milk production may well have been reduced without them undertaking any anticipatory action to reduce exothermic processes. We do not know whether they were under high heat load because the important source of heat would have been solar radiation, which was not reported.

Unfortunately, we do not have measurements of solar radiation at our field site. However high levels of solar radiation increases air temperature (this is one reason why sunny days are usually hotter than cloudy days, even in the shade), so changes in solar radiation are to a high degree correlated with the temperature data that we have measured. Furthermore, banded mongooses often forage and rest in the shade (e.g. under bushes and trees). Therefore, it cannot be assumed that banded mongooses are directly exposed to solar radiation, even if we had measured it.

Values for incoming radiant flux are available from NOAA or similar sites for every point on earth and at all times, and it is possible to predict microclimates at hourly intervals everywhere (Kearney, M.R., Gillingham, P.K., Bramer, I., Duffy, J.P. and Maclean, I.M., 2020. A method for computing hourly, historical, terrain-corrected microclimate anywhere on earth. *Methods in Ecology and Evolution*, 11(1), pp.38-43), though some assumptions have to be made about cloud cover and the nature of the terrain. The authors seem to believe that foraging and resting under trees would protect their animals from the effects of solar radiation. A recent study shows that Brazilian savanna trees reduce the effects of radiation by less than 20% (Teixeira, B.E., Nascimento, S.T., do Nascimento Mós, J.V., De Oliveira, E.M., Dos Santos, V.M., Maia, A.S.C., Fonsêca, V.D.F.C., Passos, B.M. and Murata, L.S., 2022. The potential of natural shade provided by Brazilian savanna trees for thermal comfort and carbon sink. *Science of The Total Environment*, 845, p.157324).

2. Your crucial results are presented in Table 1 and Figure 1. I believe the regression lines of Figure 1, drawn without any data points, conceal some problematic properties of your data set. For example, from the supplemental information, when rain was more than 4 mm, there were only four pups for which lactating females were exposed to maximum air temperature less than 28°C but ten for which lactating females were exposed to more than 32°C. When rain was less than 2 mm, there were 17 pups for which lactating females were exposed to maximum air temperature less than 28°C but none for which lactating females were exposed to more than 32°C. So the pups were not distributed anywhere near evenly across combinations of rainfall and maximum air temperature to which the lactating females were exposed, as valid regression analysis requires, though the figure may give the impression that they were. Further, again from the supplemental information, the mass of pups for which the lactating females were exposed to rain less than 2 mm and maximum air temperature less than 28°C was 155 ± 34 (SD) g; according to your formulation those females would have had the least food intake and been able to dissipate metabolic heat most easily. The mass of pups for which the lactating females were exposed to rain more than 4 mm and maximum air temperature more than 32°C was 149 ± 29 g; according to your formulation those females would have had the highest food intake and been able to dissipate metabolic heat least easily. But a t-test shows that those mean masses were not significantly different. So the two most-extreme combinations of temperature and rainfall produced pups of the same mass, which is hard to reconcile with your conclusion that the temperature/rain interaction was the weather factor with which changes in mass were associated.

The reviewer is correct in pointing out that pup masses were not evenly distributed across rainfall

and temperature. However, this is the case with the vast majority of biological data obtained from a wild system (which is often normally rather than uniformly distributed), and nor is it a requirement of a linear regression; regressions make no assumptions of the distribution of explanatory variables

Agreed, provided that you test for heteroskedasticity, and no such test was reported.

The idea that the combination of high temperature and high rainfall is the opposite extreme in comparison to low temperature and rainfall is incorrect.

I should have picked the high temperature/low rainfall vs low temperature/ high rainfall extremes when I was trying to assess the size of your temperature/rainfall effect, but you would have rejected a t-test there too.

3. The single measure of the status of the thermal environment that you used was the maximum air temperature over the period of exposure. But you point out (lines 106-109) that the lactating females were not active in the middle of the day, which means that they were not exposed to those maximum temperatures. The appropriate temperature to enter into your analysis, I believe, would be the temperature during the foraging bouts, say temperature for the three hours after dawn and the three hours before dusk, averaged over all the days of the exposure.

Unfortunately, we do not have long-term data available detailing the temperature during foraging bouts, only daily maximum temperature.

That is a pity when it is so easy and inexpensive to measure ambient temperatures continuously e.g. with iButtons. However, I agree with your contention that, in your context, a higher maximum temperature was a reasonable surrogate for a hotter day. I think that should say explicitly, i.e. not say that the animals were exposed to the temperatures that you measured (because they were not) but that the temperatures identified hotter and cooler days.

4. Were rainfall and maximum air temperature independent variables? I'd like to see a plot of rainfall against maximum air temperature. Also, one way in which rainfall and ambient heat load on the animals could have interacted would be if cloud cover increased in the rainier months and so reduced solar radiation. Without a measure of radiation, you can't test that possibility directly. However, I believe that you should re-run your LMM with season as a covariate. Since conventional seasons aren't relevant at the equator, I think that you should designate four seasons around rainfall, for example Season 1 = March-May, Season 2 = June-August, Season 3 = September-November, Season 4 = December- February. That re-run also should use average temperature at the time of foraging and not maximum air temperature.

Rainfall and air temperature are not strongly correlated (correlation coefficient 0.081, see the plot below). We have added this information to the manuscript (L147-148). However, given that the seasons that the reviewer suggests above are defined based solely on rainfall, season is by definition dependent on rainfall. It would therefore be analytically inappropriate and introduce extreme covariation of explanatory variables if we were to add 'season' into our model as suggested.

Thanks for running the correlation. It's valuable to have established that temperature and rainfall indeed were independent variables. And you are right about the covariation.

5. You say (lines 79-80), I believe correctly, that “Studying equatorial species can therefore allow us to decouple the impacts of temperature variation and food supply on energy dynamics”. However, studying equatorial species makes it very difficult to discover effects resulting from differences in the thermal status of the environment, because that thermal status varies so much less on the equator than it does at higher latitudes. One of the reasons for temperature not turning up as a factor on its own in your LMM might be that there just was not enough variation in temperature.

we still find a significant negative effect of high temperature on pup weight (estimate = -4.63, std error = 1.89, d.f. = 175.46, t-value = -2.46, p=0.015).

I am not surprised.

6. You have ignored evaporative cooling. Evaporative cooling is by far the most-effective avenue for dissipating body heat, and, except when ambient water-vapour pressure is too high, can dissipate the heat produced by strongly exothermic processes, including heavy exercise. Evaporative cooling certainly can dissipate, easily, the heat produced by lactation, and sustained evaporative cooling is perfectly possible in equatorial environments where water is readily available.

It seems to be a strong statement to say "Evaporative cooling certainly can dissipate, easily, the heat produced by lactation" (emphasis ours) and "your lactating females would have no difficulty dissipating body heat"

As far as I know, the energy cost of lactation in banded mongoose has not been measured and the values in mice are of little value because the mice are so much smaller. It has been measured in cotton rats, the mass of which is closer (Randolph, P.A., Randolph, J.C., Mattingly, K. and Foster, M.M., 1977. Energy costs of reproduction in the cotton rat, *Sigmodon hispidus*. *Ecology*, 58(1), pp.31-45) and is about 2 ml O₂.g⁻¹.h⁻¹. At 20 J. ml⁻¹, that is 40 J.g⁻¹.h⁻¹. At a latent heat of evaporation of 2300 J.g⁻¹, removal of all that heat of lactation would require the mongoose of mean mass (189g) to evaporate about 3 g of water per hour. Assuming that their prey is 75% water, that means that they need to consume only 4 g of prey per hour to replace the water lost.

The resting metabolic rate of non-lactating cotton rats also is about 2 ml O₂.g⁻¹.h⁻¹ so the water in another 4 g.h⁻¹ of prey would dissipate resting metabolic heat plus lactational heat. I do not think that my statement is unnecessarily strong.

I would not expect the banded mongoose to pant except perhaps *in extremis*. Modern research on small mammals, especially by Cooper and Withers, has shown that many small mammals use percutaneous evaporative cooling, without sweat glands.

I think that your text requires an explicit statement that your conclusions about heat dissipation rest on an assumption that banded mongoose do not employ evaporative cooling. That means, of course, that they cannot survive if ambient heat (radiation plus convection) imposes a load on the body that causes body temperature to rise.

Line by line comments

8.Line 28: equatorial species experience similar temperatures year-round, but those living at high altitude on the equator do not experience high temperatures year-round.

9.Line 44: there are many other strategies that fall under the rubric of “behavioural thermoregulation”.

This is indeed the case. In our introduction, we provide two examples of behavioural thermoregulation; reduced activity and microhabitat selection. These are clearly listed as examples and are not presented as an exhaustive list.

You said “these strategies, collectively called ‘behavioural thermoregulation’”, rather than “these strategies, which are examples of ‘behavioural thermoregulation’”

10.Line 45: ceasing foraging in hot periods does not necessarily reduce energy intake. See Hetem, R.S. et al., 2012. Activity re-assignment and microclimate selection of free-living Arabian oryx: responses that could minimise the effects of climate change on homeostasis? *Zoology*, 115(6), pp.411-416.

11.Line 71: again, not at high altitude.

12.Line 75: you need to make clear that, in temperate environments, increasing temperature is associated with increasing food, because in arid environments, increasing temperature is associated with less food.

13.Line 81: against what alternative or alternatives did you test HDL theory? I do not believe that you were testing the theory. You were seeking support for it.

We test HDL theory against the possibility that energy intake alone influences pup growth (via influencing energy available for the production of milk).

But you did not explore the more crucial alternative that heat stress leading to maternal hyperthermia compromises lactation.

14.Line 86: underground rearing buffers the pups much more against the effects of solar radiation than it buffers then against high air temperature.

While we have no data on the temperatures of banded mongoose dens (access is very difficult), studies on other species existing at similar temperature ranges have demonstrated substantial buffering of temperature.

I agree about the buffering, but studies on other species also have shown that use of burrows by diurnally active small mammals has more to do with escaping radiation.

15.Line 87: what underground rearing makes possible is separating thermal effects on pups (see van de Ven et al., 2020. Effects of climate change on pup growth and survival in a cooperative mammal, the meerkat. *Functional Ecology*, 34(1), pp.194-202) from thermal effects on lactating females.

16. Line 90: reduced time for foraging is a problem only if it requires foraging all day to meet the daily energy requirement. Does it require foraging all day, for banded mongoose in equatorial habitats?

Our preliminary data finds that foraging time and activity levels are reduced in high ambient temperatures (see our response to reviewer 3 below), however we do not have sufficient data on food intake to determine whether lactating females struggle to meet their daily energy needs when their foraging time is reduced (food intake is not easy to determine in a wild system).

I look forward to seeing those results.

17. Line 94: you would make the same prediction if high ambient heat load compromised the physiology of lactation.

The HDL theory proposes that high ambient temperatures do indeed compromise lactation, so this is not an alternative hypothesis (see our earlier responses).

I agree, but HDL says that lactating dams reduce lactation to avoid hyperthermia. That's different from heat stress compromising the physiology of lactation because the dams become hyperthermic.

18. Line 97-99: the implication is that food density was inadequate when rainfall was low, even in your equatorial habitat. What independent evidence do you have that the lactating females couldn't meet their energy needs fully even when rainfall was low?

our data strongly suggests that rainfall over the prior 30 day period limits the energy intake of female banded mongooses.

Hopefully your new study will provide the data, because if food is sufficient at low rainfall, it is difficult to justify your conclusions.

19. Line 113: were there any heatwaves in your study period?

changed the term 'heatwave' for 'hot conditions'

I would prefer 'hotter conditions'. Across the spectrum of environments to which diurnal animals are exposed, 32C air temperature isn't particularly hot.

20. Line 115: more abundant at higher rainfall doesn't necessarily mean inadequate at lower rainfall.

21. Line 117: what other variables did the weather station measure? You have not reported any. Did you have access to any measure of radiation, water vapour pressure or wind speed? Convective heat loss could have been higher at the higher air temperatures than at the lower ones if wind speed increased sufficiently at times when air temperature was high. So, depending on the wind, lactating females could have dissipated heat to the air more easily at the higher air temperatures.

22. Line 120-122: describe your "methods" in the past tense. You are telling us what you have done, not what you are doing.

23. Line 128: when the pups were caught by hand, you had an ideal opportunity to measure their rectal temperatures, which would have allowed you to test pup body temperature as a

covariate, and to eliminate any febrile pups, which was important because fever induces anorexia.

In order to avoid dishabituating the animals and causing undue stress, we do not manipulate the pups during handling,

I think that the major stress would have come from the act of catching the pups. If their genitals were exposed, so presumably was the anus, and I think that you could have measured temperature within 30 s with a fine-wire thermocouple. Those data would have been unique and invaluable. You also could have obtained a surface temperature of the underbelly instantaneously with an infrared thermometer.

24.Line 132: if the balance accuracy was 1.5 g, how could you measure mass to a tenth of a gram and its SE to a hundredth of a gram? Also, SD not SE is the appropriate statistic here, because you are concerned with variance within the population not between populations.

25.Line 146: what was the accuracy of the thermometer in the weather station? Giving air temperatures to the second decimal place implies that the thermometers were accurate to 1/100°C. In the supplemental information, you gave some temperatures to eight decimal places, when your thermometer probably wasn't accurate even to one decimal place.

Raw temperature data were measured to 0.1 degree, which we have now explained in the main text (L111). Here, we present mean values rather than raw data, and there is no requirement for summary statistics to be presented at the same precision as the data. Nevertheless, we have now changed all values in the main text to 1dp at your suggestion. In the dataframe, we retain mean values as they are, as reducing the number of decimal places will not qualitatively influence our results.

Your raw temperature was measured to a resolution of 0.1 degree, not an accuracy of 0.1 degree; you do not say that you calibrated the thermometer. Nevertheless, you are correct that, with sufficient measurements, the mean can be more precise than individual measurements.

I'll leave the decision about the supplemental information to the editor. I wouldn't like my journal to be cluttered with meaningless numbers.

26.Line 146: again, giving rainfall to two decimal places implies that the rain gauges were accurate to 1/100 mm.

27.Line 146-149: I think that the appropriate covariate would have been ratio of number of pups to number of lactating females.

We agree that this would be a more appropriate covariate. However, as pups are born underground and remain there for the first ~30 days of their life, we are unable to count the number of pups at birth, and it is likely that some pups die before we are able to observe them. Due to the resulting uncertainty in pup:female ratios at birth, we therefore include in our model the number of lactating females.

I had in mind ratio of lactating females to pups at emergence, which I think is a more appropriate covariant than raw number of lactating females, though what you really would like is the ratio at birth.

28.Line 175, Figure 1: were the slopes of the regression lines significantly different to zero? If they were, why would pup mass decrease with increasing prior rainfall at the highest temperature? According to your postulates, that trend implies less maternal food intake at the higher rainfall. Why? Or is there another factor at play?

29.Line 175, Figure 1: though I think that air temperature was relatively unimportant both in determining heat load (radiation would have much more important) on the lactating females, and in determining their capacity to dissipate body heat (water vapour pressure would have been much more important), if you want to make an argument based on air temperature, you need to show that the sets of temperatures represented by your three means were different statistically.

30.Line 176: again, implying that you could measure air temperature to an accuracy of 0.01°C.

31.Line 181: only if the thermoregulation of the lactating animals was compromised by the higher air temperature. We don't know what their surface temperature was (it can be measured remotely by thermography) but it would have been high when they were in the sun, perhaps even higher than deep-body temperature. Let's say 40°C. Then the gradient for convective heat loss was $40 - 28.3 = 11.7^\circ\text{C}$ at your lowest temperature and $40 - 30.9 = 9.1^\circ\text{C}$ at your highest temperature. So convective heat loss would have differed by only about 20%, and that assumes that wind speed was the same at all air temperatures. Even if the animals weren't cooling by evaporation, that's a small difference. To be consistent with HDL theory, you would have to say that the lactating females were "forced to suppress exothermic processes" (line 183, i.e. to reduce their lactation sufficiently to prevent the pups growing) because they "knew" that their cooling by convection was going to be reduced by 20%, an amount easily compensable by evaporative cooling.

the HDL hypothesis is not about predicting future conditions.

I think that it is, and so, I believe, do Speakman and Król.

our findings are consistent with HDL theory (rather than stating that they are a result of HDL).

More accurately stated, I think.

32.Line 189-190: so, you are saying that air temperatures reaching a maximum of 30.9°C outside the den might have imposed a risk of harmful hyperthermia on pups in the den. In adult slender mongoose in the lab, rectal temperature was elevated but held constant when the air temperature was 38°C permanently (Kamau et al. 1979).

In both cases the authors specifically state that body (rectal) temperatures continued increasing at higher temperatures, in contrast to the reviewer's claim.

I said at 38C, not 43C

33.Line 192: there would have been benefit only when the pups were in the den. As soon as they were outside, and under radiant heat load, a high surface area to mass ratio would have been a disadvantage. The larger pups would have been better off outside the den.

34.Line 199: Van de Ven et al. 2020 is relevant to this paragraph.

35.Line 209: observing pups in the den may have been beyond your resources, but It certainly was possible, with fibre-optic cameras.

Over the 20+ years of our study, we have indeed tried to observe pups in the den using such cameras, however due to the positioning of dens (often in inaccessible locations such as dense bushes or underneath buildings), we have been unable to reliably collect such observations.

I can imagine how frustrating that must have been because you could have added so much more by seeing the pups.

36.Line 212: I don't think that the comparisons discussed in this paragraph are relevant. Your pups did not experience high temperatures in early ex-utero development.

37.Line 222-224: your analysis did not confirm an effect of temperature, but only of a temperature/rainfall interaction.

By definition, a variable involved in an interaction has an effect.

You are correct. Just not an independent effect, although your subsequent analysis indeed did reveal an independent effect.

38.Line 224: I do not understand why would you expect high food abundance to compensate for high ambient heat load? High food abundance presumable would lead to more milk consumption and higher metabolic heat production. If you are working within HDL theory, how does that compensate for high ambient heat load?

Appendix D

Dear Editor,

Thank you and the reviewers for your constructive comments and for the opportunity to resubmit our manuscript. We have addressed all comments and detail our responses to each individual point below. We hope the paper now meets your requirements for publication and look forward to hearing back from you in due course.

Kind regards,

Dr Hazel Nichols & co-authors

Handling Editor's Comments to Authors:

The current contribution received mixed responses from the reviewers. Please revise your paper taking cognizance of the points brought to your attention in the review process. Reviewer 1 has a number of valid points, please try to address these in the revision where possible. I would appreciate a detailed covering letter highlighting how you have taken on board the points raised in the review process. Please use the MS word track changes option to show the corrections you have made in the manuscript.

Thank you for your thoughtful comments and guidance. We have addressed all of the points raised by the reviewers below.

Board Member's comments:

This paper is investigating the heat dissipation limit theory (HDL) under field conditions in banded mongoose. Similar to reviewer 2 and 3, I thoroughly enjoyed reading this paper and believe that it will make an important contribution to the field.

We are glad that you regard our research on a wild species as making an important contribution to the field.

However, I also encourage the authors to at least integrate part of the reviewer's feedback, such as considerations on whether their results could have in fact been caused by maternal hypothermia, into the discussion.

Thank you for this comment. This distinction in physiological mechanism does not appear to be a focus of the originators of the HDL hypothesis (see our response to reviewer 2 below). Nevertheless, it is an interesting distinction that we think would benefit from additional exploration in future studies and we now include that 'it is unclear from these studies whether females stop lactating before or after hyperthermia sets in' L58-59 and 'lactating females may be forced to suppress exothermic processes such as milk production (either at or approaching their critical thermal maximum) in order to avoid hyperthermia' in the discussion, L176.

As an additional minor comment, I also recommend clarifying the use of “high temperature” in the introduction as this could mean both, ‘high body temperature’ or ‘high air/ambient temperature’.

Thank you for highlighting this potential area of misunderstanding. To clarify, we have changed the title to ‘high ambient temperature’ and have also clarified that we are referring to ambient temperatures in the abstract (L24) and methods (L112). This point is also included in the first paragraph of the discussion (L175). Hopefully reaffirming this point throughout will avoid any confusion.

Referee: 1

Comments to the Author(s)

great job on a challenging problem. I thoroughly enjoyed not only the manuscript but your responses to all of us.

Thank you for your kind and positive comments, we are glad you enjoyed our work.

Referee: 3

Comments to the Author(s)

The Authors have revised the manuscript sufficiently to be considered for publication in Biology Letters. Some minor suggestions and editorial changes are listed below.

1) Title: it would be beneficial to add ‘ambient’ or ‘air’ temperature, to distinguish it from ‘body’ or ‘den’ temperature. Furthermore, ‘offspring body mass’ may be more accurate, in case ‘offspring mass’ is understood as ‘litter mass’, i.e., mass of all pups per den.

We have now changed the title to “Small increases in ambient temperature reduce offspring body mass in an equatorial mammal”. Following the board member’s advice, we have also clarified this throughout the manuscript.

2) Abstract & Results/Discussion: it would be important to present the Reader with some magnitude of changes in T_a and pup body mass. How small are the changes in T_a to have an impact on pup body mass? How big are the changes in pup body mass?

This is difficult to put numbers on because temperature is involved in an interaction with rainfall, i.e. the accurate answers to the questions posed are ‘it depends on rainfall’. However, Figure 1B displays the influence of rainfall at the mean temperature and at 1 standard deviation above and below this (28.3 and 30.9°C respectively). As illustrated in the figure (and because our study site is at the equator) absolute variation in mean maximum temperature is relatively small 27.0-32.1°C; see also lines 216-217 in the Discussion). Again, the numbers requested (How small are the changes in T_a to have an impact on pup body mass, and how big are the changes in pup body mass) depend heavily on rainfall because of the interaction, from no predicted change in pup mass at these three temperatures at ~2mm

rain to ~40% increase in body mass (from ~170g to ~240g) between the highest and lowest temperature values shown at ~6.5mm rain, which differ by only ~2°C. Due to the effect of the interaction, we believe that choosing a particular value to illustrate these numbers in the text would be potentially misleading. However, our Figure enables readers to quantify these issues while also conveying the complexity of the scenario.

3) Abstract: Unclear what the Authors mean by the 'opportunity costs'? This term is never discussed in the main part of the paper.

For the purposes of our paper, 'opportunity costs' refer to reduced foraging opportunities, which we explain in the abstract from lines 17-19 and also lines 43-46 in the Introduction. We have now added further clarification into the introduction by stating that "...behavioural thermoregulation', can be costly in terms of lost opportunities..."(lines 43-46).

4) Figure Legend: unclear how it is evident from CIs that there no effects of rainfall on pup mass at higher Ta.

Looking at the effect of rainfall at a higher Ta (30.86°C), the gradient of the line is shallow and confidence intervals easily encompass a flat line (i.e. no effect). In contrast, the gradient of the line for the lowest temperature (28.30°C) is steep and the CIs are such that there is no way to draw a flat horizontal line completely within the confidence intervals, indicating an increase in body mass as rainfall increases.

5) Number and unit should be separated by space. Hyphens, en dashes and em dashes mixed.

The submission guidelines for Biology Letters state: "Note that we do not have strict formatting requirements in terms of e.g. fonts, spacing, spelling etc. ... Our typesetters will set your manuscript into house style after acceptance." In accordance with this we will allow the typesetters to format in their preferred style. However, if the editors would like us to make any changes prior to typesetting, we will be happy to make any such changes.

For authors, round 2

Thank you for the attention that you have given to my comments. You have resolved several of the problems that I had. The comments that follow relate mostly to those that I regard as unresolved or not fully resolved.

We are glad that our response clarified many of the issues raised by this reviewer.

.....

It is whether pup mass was associated with the weather to which the lactating females were

exposed that you set out to analyse. You concluded, using LMM analysis, that, on their own, neither the maximum air temperature over the period of lactation nor the rainfall at a time earlier, to take account of latency in consequences for prey abundance, had any significant effect on pup mass, but the interaction between maximum temperature and earlier rainfall indeed did so.

Note that we do not state that the environmental variables that we measured have no significant impact on pup mass alone. Our prediction based on HDL theory was that we would find an interaction between temperature and rainfall on pup mass, and this interaction is what we tested for. As both variables are involved in a significant interaction, we do not report significance values for their individual effects on pup mass as the parameters for these main effects are unintuitive and often biologically misleading in such cases.

A relationship between pup mass and temperature alone is intuitive to me if lactation fails under heat stress. To make your approach clear, I think that you should add a sentence, in an appropriate place, to the effect that “we did not demonstrate, and nor did we seek to test, a relationship between pup mass and either temperature or rainfall as independent variables”.

Note that our approach would have tested for independent effects of temperature and rainfall if these effects were indeed independent; if no evidence for an interaction was found then the independent effects would be what our model would have reported (so we cannot state that we did not seek to test these relationships). However, the fact is that our results did find evidence for an interaction and therefore we can conclude that these effects are not independent.

Unfortunately, in my opinion, you were so committed to finding support for the heat dissipation limit (HDL) theory of Speakman and Król (reference #9) that you failed to address the data before you on its merits and you overlooked many deficiencies in the arguments that you used to reach your conclusions. In my view, I do not believe there was any solid evidence provided for factors contributing to variability in pup mass at emergence other than age. Also in my view, HDL theory warranted no more than a paragraph in the “discussion”.

We had no prior commitments to finding support for the HDL theory. We present an alternative possibility in the introduction (that pup mass is related to the energy available to lactating females via resource availability) that we also test. In fact, our introduction begins with describing this alternative to HDL. Our statistical model demonstrates that temperature and rainfall are associated with pup mass (not only that pup mass increases with age).

That is not the alternative hypothesis that I would have liked to have seen. Given the extensive literature on heat stress and lactation, my alternative hypothesis would have been that lactation is compromised by heat stress leading to maternal hyperthermia, irrespective of food supply. The problem is that you don't have a good measure of heat stress. Your LMM indeed shows that the interaction between temperature and rainfall is associated statistically with pup mass at emergence, but I submit that the effect is weak, as would have been more evident if you have plotted your Fig. 1 over the full range of pup masses (93 -307 g), rather than about 150-250 g.

We appreciate that the reviewer's choice of alternative hypothesis would be different from ours, however we maintain that our choice is appropriate for our wild study system. The alternative hypothesis that lactation stops when maternal hyperthermia sets in (rather than stops to prevent hyperthermia from setting in) is not possible to test without detailed

measurements of lactation and maternal temperature. These measurements would be best conducted under highly controlled laboratory conditions, rather than in our wild study system, and we would welcome such studies in the future to reveal the details of the mechanism.

Note that our choice of alternate hypothesis is consistent with Speakman and Kroll who originated the HDL hypothesis. In their 2010 review (*Journal of Animal Ecology*, 79, 726–746) these authors clarified both their theory and the alternatives against which it should be tested as follows: “The HDL theory suggests that ability to lose heat is the key process that controls maximal energy expenditure. The heat dissipation capacity, rather than the ability to acquire resources from the environment, provides the boundary condition that leads to trade-offs in animal energy allocation.”

Our study tests this alternative hypothesis that food availability determines lactation and therefore pup growth. We believe that this is an important hypothesis to test and also generates meaningful predictions in our wild study system. Studying the impact of ambient temperature and rainfall on pup growth in a wild system also provides important information in regard to the potential impacts of heatwaves and climate change, regardless of the details of the mechanism through which they act.

General comments

1. Animals undoubtedly can be exposed to environments in which they have difficulty dissipating body heat. That’s simply a consequence of the laws of physics, not a consequence of HDL theory. Those environments typically will have high radiant heat load and high water-vapour pressure, inhibiting evaporative cooling. There is no information about either radiation or water-vapour pressure in your paper. In such environments, exothermic processes may well cause detrimental hyperthermia (not all hyperthermia is detrimental), and that hyperthermia may reduce or eliminate the exothermic process. Runners who get too hot have to stop running. That’s a feedback process. What’s different about HDL theory is that it proposes a feedforward process. HDL proposes that animals, not yet hyperthermic, receive some kind of warning signal about impending thermal threats and reduce exothermic processes in anticipation of running into trouble. So, to claim that reducing an exothermic process like lactation is consistent with HDL, you have to demonstrate that the reduction is anticipatory, a demonstration for which I don’t think that your data are suitable.

Both our understanding (which reviewers 1 and 3 appear to concur with) and that of the published literature of the topic (including from the originators of the hypothesis) is that the HDL theory does not propose that animals ‘receive some kind of warning signal’ about impending thermal threats. It instead proposes that the ability to perform exothermic processes (such as milk production) is limited by their ability to dissipate body heat because of the temperatures they are experiencing at the time.

Then what is different between HDL theory and the theory, for which there is plenty of evidence, some of which you cite, that lactation fails under heat stress leading to maternal hyperthermia? I would not regard runners stopping running because they are too hot as an example of HDL in operation.

The key word in the quotation from the original Speakman and Król paper is “avoid”: “*upper boundary on total energy expenditure is imposed by the maximal capacity to dissipate body heat and therefore avoid the detrimental consequences of hyperthermia*”. How do you “avoid” without preemptive

action? Similarly, in a later quotation: “*because ingesting additional food would have made them dangerously hyperthermic*”. Not ingesting additional food in anticipation of possible hyperthermia is a feedforward process.

The authors cite one of Speakman’s elegant shaving experiments. There is another Speakman shaving experiment on voles (Simons, M.J., Reimert, I., van der Vinne, V., Hambly, C., Vaanholt, L.M., Speakman, J.R. and Gerkema, M.P., 2011. Ambient temperature shapes reproductive output during pregnancy and lactation in the common vole (*Microtus arvalis*): a test of the heat dissipation limit theory. *Journal of Experimental Biology*, 214(1), pp.38-49) which concludes “*Shaving fur off dams at 30°C resulted in faster growth of pups; however, no significant increase in food intake and or milk production was detected*”.

Note that in all of these cases, and in the experiment that the reviewer mentions here, the animals are responding to ambient temperatures that they are currently experiencing, not potential future temperatures as the reviewer proposed in their initial comment. We agree with the reviewer that the possibilities that lactation fails under hyperthermia versus lactation reducing to prevent hyperthermia are not clearly differentiated from each other in the literature. This is reflected in the literature describing HDL, e.g. Speakman & Krol (2010, *Journal of Animal Ecology*, 79, 726–746) in which they discuss the origin of the hypothesis, and in the experiments that were designed to test HDL theory; animals were placed in different ambient temperatures or had their fur shaved, rather than measuring body temperatures and assessing whether lactation reduced immediately prior to or after the onset of hyperthermia. Zhao et al (2020, *PNAS*, 117 (39) 24352-24358) investigate lactation and pup mass in mice over a similar ambient temperature range to our study (21–36°C). They found a substantial reduction in lactation (31.8%) at 27 compared to 21°C, and lactation reduced further (by 61.9%) at 30°C and (by 82.1%) 33°C. Here, lactation reduced gradually as ambient temperature increased, rather than dropping off sharply at a hard threshold as females became hyperthermic, suggesting that females reduce lactation to avoid hyperthermia rather than because of it. Females kept at ambient temperatures of 36°C displayed symptoms of hyperthermia and some died after 24 hours. In contrast, lactating females kept at lower temperatures didn’t display symptoms of hyperthermia and were kept at these temperatures for at least 14 days, suggesting that the reduction in lactation was not due to hyperthermia.

Based on the originators of the HDL hypothesis not making a clear distinction between these two possibilities, we do not see strong evidence that the HDL hypothesis is contingent on this distinction. However, as the reviewer and board member point out, this distinction is worth considering in the future, and could be determined using laboratory studies. Future reviews of the area may also benefit from discussing in detail and clarifying the terminology and proposed mechanisms of the HDL hypothesis, but this is outside the scope of our study.

2. It has been known for at least 70 years that high ambient heat load compromises the physiology of lactation, potentially resulting in massive reduction of milk production, especially in high-lactating animals. See the diagram for cows, from Hafez, E.S.E., 1968.

Adaptation of domestic animals, based on 1950 data :

If your lactating females were under high enough heat load, their milk production may well have been reduced without them undertaking any anticipatory action to reduce exothermic processes. We do not know whether they were under high heat load because the important source of heat would have been solar radiation, which was not reported.

Unfortunately, we do not have measurements of solar radiation at our field site. However high levels of solar radiation increases air temperature (this is one reason why sunny days are usually hotter than cloudy days, even in the shade), so changes in solar radiation are to a high degree correlated with the temperature data that we have measured. Furthermore, banded mongooses often forage and rest in the shade (e.g. under bushes and trees). Therefore, it cannot be assumed that banded mongooses are directly exposed to solar radiation, even if we had measured it.

Values for incoming radiant flux are available from NOAA or similar sites for every point on earth and at all times, and it is possible to predict microclimates at hourly intervals everywhere (Kearney, M.R., Gillingham, P.K., Bramer, I., Duffy, J.P. and Maclean, I.M., 2020. A method for computing hourly, historical, terrain-corrected microclimate anywhere on earth. *Methods in Ecology and Evolution*, 11(1), pp.38-43), though some assumptions have to be made about cloud cover and the nature of the terrain. The authors seem to believe that foraging and resting under trees would protect their animals from the effects of solar radiation. A recent study shows that Brazilian savanna trees reduce the effects of radiation by less than 20% (Teixeira, B.E., Nascimento, S.T., do Nascimento Mós, J.V., De Oliveira, E.M., Dos Santos, V.M., Maia, A.S.C., Fonsêca, V.D.F.C., Passos, B.M. and Murata, L.S., 2022. The potential of natural shade provided by Brazilian savanna trees for thermal comfort and carbon sink. *Science of The Total Environment*, 845, p.157324).

Our study animals spend a considerable amount of time in the shade of trees and bushes (which provide dappled shade) and also under more solid objects such as buildings, rocks, logs and (occasionally) in their underground dens. We would therefore not like to assume that this has little impact on thermoregulation in our current study. Furthermore, given that we find an association between ambient temperature and pup weight, the absence of solar radiation data does not invalidate our results regarding temperature. Nevertheless, we appreciate the suggestions and will incorporate solar radiation into future studies at our field site where feasible (ideally by measuring the solar radiation that reaches foraging mongooses).

2. Your crucial results are presented in Table 1 and Figure 1. I believe the regression lines of Figure 1, drawn without any data points, conceal some problematic properties of your data set. For example, from the supplemental information, when rain was more than 4 mm, there were only four pups for which lactating females were exposed to maximum air temperature less than 28°C but ten for which lactating females were exposed to more than 32°C. When rain was less than 2 mm, there were 17 pups for which lactating females were exposed to maximum air temperature less than 28°C but none for which lactating females were exposed to more than 32°C. So the pups were not distributed anywhere near evenly across combinations of rainfall and maximum air temperature to which the lactating females were exposed, as valid regression analysis requires, though the figure may give the impression that they were. Further, again from the supplemental information, the mass of pups for which the lactating females were exposed to rain less than 2 mm and maximum air temperature less than 28°C was 155 ± 34 (SD) g; according to your formulation those females would have had the least food intake and been able to dissipate metabolic heat most easily. The mass of pups for which the lactating females were exposed to rain more than 4 mm and maximum air temperature more than 32°C was 149 ± 29 g; according to your formulation those females would have had the highest food intake and been able to dissipate metabolic heat least easily. But a t-test shows that those mean masses were not significantly different. So the two most-extreme combinations of temperature and rainfall produced pups of the same mass, which is hard to reconcile with your

conclusion that the temperature/rain interaction was the weather factor with which changes in mass were associated.

The reviewer is correct in pointing out that pup masses were not evenly distributed across rainfall and temperature. However, this is the case with the vast majority of biological data obtained from a wild system (which is often normally rather than uniformly distributed), and nor is it a requirement of a linear regression; regressions make no assumptions of the distribution of explanatory variables

Agreed, provided that you test for heteroskedasticity, and no such test was reported.

We found no evidence of heteroskedasticity. We have now referred to standard model checks that were carried out in Line 149.

The idea that the combination of high temperature and high rainfall is the opposite extreme in comparison to low temperature and rainfall is incorrect.

I should have picked the high temperature/low rainfall vs low temperature/ high rainfall extremes when I was trying to assess the size of your temperature/rainfall effect, but you would have rejected a t-test there too.

Indeed, we feel our model choice is more appropriate than a t-test for the reasons previously explained.

3. The single measure of the status of the thermal environment that you used was the maximum air temperature over the period of exposure. But you point out (lines 106-109) that the lactating females were not active in the middle of the day, which means that they were not exposed to those maximum temperatures. The appropriate temperature to enter into your analysis, I believe, would be the temperature during the foraging bouts, say temperature for the three hours after dawn and the three hours before dusk, averaged over all the days of the exposure.

Unfortunately, we do not have long-term data available detailing the temperature during foraging bouts, only daily maximum temperature.

That is a pity when it is so easy and inexpensive to measure ambient temperatures continuously e.g. with iButtons. However, I agree with your contention that, in your context, a higher maximum temperature was a reasonable surrogate for a hotter day. I think that should say explicitly, i.e. not say that the animals were exposed to the temperatures that you measured (because they were not) but that the temperatures identified hotter and cooler days.

Adult banded mongooses are very likely exposed to maximum ambient (shade) temperatures during the hottest part of the day. To avoid these temperatures, they would have to remain below ground in their dens for extended time periods, which we rarely observe. Note that we do not explicitly state in the manuscript that banded mongooses are directly exposed to the temperatures that we measured. Based on the concise nature of Biology Letters, we do not feel that the manuscript would benefit from the addition of an explanation that the higher and lower temperatures that we measured identify hotter and cooler days (particularly as we analysed average Tmax over 30 days, rather than a finer scale measure).

4. Were rainfall and maximum air temperature independent variables? I'd like to see a plot of rainfall against maximum air temperature. Also, one way in which rainfall and ambient heat load on the animals could have interacted would be if cloud cover increased in the rainier months and so reduced solar radiation. Without a measure of radiation, you can't test that possibility directly. However, I believe that you should re-run your LMM with season as a covariate. Since conventional seasons aren't relevant at the equator, I think that you should designate four seasons around rainfall, for example Season 1 = March-May, Season 2 = June-August, Season 3 = September-November, Season 4 = December- February. That re-run also should use average temperature at the time of foraging and not maximum air temperature.

Rainfall and air temperature are not strongly correlated (correlation coefficient 0.081, see the plot below). We have added this information to the manuscript (L147-148). However, given that the seasons that the reviewer suggests above are defined based solely on rainfall, season is by definition dependent on rainfall. It would therefore be analytically inappropriate and introduce extreme covariation of explanatory variables if we were to add 'season' into our model as suggested.

Thanks for running the correlation. It's valuable to have established that temperature and rainfall indeed were independent variables. And you are right about the covariation.

I'm glad that the reviewer finds the inclusion of this information useful; we hope that readers are also reassured by this.

5. You say (lines 79-80), I believe correctly, that "Studying equatorial species can therefore allow us to decouple the impacts of temperature variation and food supply on energy dynamics". However, studying equatorial species makes it very difficult to discover effects resulting from differences in the thermal status of the environment, because that thermal status varies so much less on the equator than it does at higher latitudes. One of the reasons for temperature not turning up as a factor on its own in your LMM might be that there just was not enough variation in temperature.

we still find a significant negative effect of high temperature on pup weight (estimate = -4.63, std error = 1.89, d.f. = 175.46, t-value = -2.46, p=0.015).

I am not surprised.

6. You have ignored evaporative cooling. Evaporative cooling is by far the most effective avenue for dissipating body heat, and, except when ambient water-vapour pressure is too high, can dissipate the heat produced by strongly exothermic processes, including heavy exercise. Evaporative cooling certainly can dissipate, easily, the heat produced by lactation, and sustained evaporative cooling is perfectly possible in equatorial environments where water is readily available.

It seems to be a strong statement to say "Evaporative cooling certainly can dissipate, easily, the heat produced by lactation" (emphasis ours) and "your lactating females would have no difficulty dissipating body heat"

As far as I know, the energy cost of lactation in banded mongoose has not been measured and the values in mice are of little value because the mice are so much smaller. It has been measured in cotton rats, the mass of which is closer (Randolph, P.A., Randolph, J.C., Mattingly, K. and Foster, M.M., 1977. Energy costs of reproduction in the cotton rat, *Sigmodon hispidus*. *Ecology*, 58(1), pp.31-45)

and is about $2 \text{ ml O}_2 \cdot \text{g}^{-1} \cdot \text{h}^{-1}$. At $20 \text{ J} \cdot \text{ml}^{-1}$, that is $40 \text{ J} \cdot \text{g}^{-1} \cdot \text{h}^{-1}$. At a latent heat of evaporation of $2300 \text{ J} \cdot \text{g}^{-1}$, removal of all that heat of lactation would require the mongoose of mean mass (189g) to evaporate about 3 g of water per hour. Assuming that their prey is 75% water, that means that they need to consume only 4 g of prey per hour to replace the water lost.

The resting metabolic rate of non-lactating cotton rats also is about $2 \text{ ml O}_2 \cdot \text{g}^{-1} \cdot \text{h}^{-1}$ so the water in another 4 g.h⁻¹ of prey would dissipate resting metabolic heat plus lactational heat. I do not think that my statement is unnecessarily strong.

I would not expect the banded mongoose to pant except perhaps *in extremis*. Modern research on small mammals, especially by Cooper and Withers, has shown that many small mammals use percutaneous evaporative cooling, without sweat glands.

I think that your text requires an explicit statement that your conclusions about heat dissipation rest on an assumption that banded mongoose do not employ evaporative cooling. That means, of course, that they cannot survive if ambient heat (radiation plus convection) imposes a load on the body that causes body temperature to rise.

Given that we do not know the degree to which evaporative cooling is used in banded mongooses, we prefer not to speculate that it is sufficient to dissipate the heat produced by lactation, or that our conclusions rely on a lack of evaporative cooling. Furthermore (as the reviewer acknowledges), a reduction in lactation at higher temperatures has been found in studies from species as small as mice to as large as cows (mostly under controlled/laboratory conditions). This would seem unlikely to be the case if evaporative cooling certainly could dissipate the heat produced by lactation. Furthermore, we find that pup mass is reduced under higher temperatures, which is consistent with the idea that high temperatures constrain lactation.

Line by line comments

8.Line 28: equatorial species experience similar temperatures year-round, but those living at high altitude on the equator do not experience high temperatures year-round.

Yes, which is why we used the word “often” in this sentence. In the introduction, we clarify that this applies to species at low latitude, but we want to keep the abstract as concise as possible.

9.Line 44: there are many other strategies that fall under the rubric of “behavioural thermoregulation”.

This is indeed the case. In our introduction, we provide two examples of behavioural thermoregulation; reduced activity and microhabitat selection. These are clearly listed as examples and are not presented as an exhaustive list.

You said “these strategies, collectively called ‘behavioural thermoregulation’”, rather than “these strategies, which are examples of ‘behavioural thermoregulation’”

By ‘these strategies’, we are referring to the changes in behaviour performed to avoid hyperthermia. We believe this is clear once read in context, and we feel that the current sentence structure is more concise:

“Many species attempt to avoid hyperthermia by changing their behaviour, for example through a reduction in activity or microhabitat selection of cooler locations (6,7). However, these strategies, collectively called ‘behavioural thermoregulation’ ...”

10.Line 45: ceasing foraging in hot periods does not necessarily reduce energy intake. See Hetem, R.S. et al., 2012. Activity re-assignment and microclimate selection of free-living Arabian oryx: responses that could minimise the effects of climate change on homeostasis?*Zoology*, 115(6), pp.411-416.

Here we state that they can reduce energy intake, for example through ceasing foraging. We do not state that this is always the case. As *Biology Letters* publishes concise papers, it is not possible to explore the potential exceptions to all statements made; rather we aim to indicate the broad pattern that ceasing foraging can lead to reduced energy intake.

11.Line 71: again, not at high altitude.

We clearly state on line 72 that we refer to low altitudes.

12.Line 75: you need to make clear that, in temperate environments, increasing temperature is associated with increasing food, because in arid environments, increasing temperature is associated with less food.

We clearly state this (Line 76-77).

13.Line 81: against what alternative or alternatives did you test HDL theory? I do not believe that you were testing the theory. You were seeking support for it.

We test HDL theory against the possibility that energy intake alone influences pup growth (via influencing energy available for the production of milk).

But you did not explore the more crucial alternative that heat stress leading to maternal hyperthermia compromises lactation.

The reviewer is correct that we did not explore this hypothesis. As explained previously, we believe that this hypothesis would be better addressed by a controlled laboratory experiment. Our study aims to address the potential impacts of increasing temperatures on a wild system, rather than determine the intricacies of the physiological mechanism.

14.Line 86: underground rearing buffers the pups much more against the effects of solar radiation than it buffers them against high air temperature.

While we have no data on the temperatures of banded mongoose dens (access is very difficult), studies on other species existing at similar temperature ranges have demonstrated substantial buffering of temperature.

I agree about the buffering, but studies on other species also have shown that use of burrows by diurnally active small mammals has more to do with escaping radiation.

We are glad that you agree with our point regarding buffering.

16. Line 90: reduced time for foraging is a problem only if it requires foraging all day to meet the daily energy requirement. Does it require foraging all day, for banded mongoose in

equatorial habitats?

Our preliminary data finds that foraging time and activity levels are reduced in high ambient temperatures (see our response to reviewer 3 below), however we do not have sufficient data on food intake to determine whether lactating females struggle to meet their daily energy needs when their foraging time is reduced (food intake is not easy to determine in a wild system).

I look forward to seeing those results.

17. Line 94: you would make the same prediction if high ambient heat load compromised the physiology of lactation.

The HDL theory proposes that high ambient temperatures do indeed compromise lactation, so this is not an alternative hypothesis (see our earlier responses).

I agree, but HDL says that lactating dams reduce lactation to avoid hyperthermia. That's different from heat stress compromising the physiology of lactation because the dams become hyperthermic.

The studies conducted by those who proposed the HDL theory to test this theory do not make a distinction between these two possibilities. We agree that the details of the mechanism would benefit from clarification but propose that a future laboratory study would better differentiate between these possibilities. We have added into the introduction (L58-59) that it is currently unclear whether lactation stops before or after hyperthermia sets in.

18. Line 97-99: the implication is that food density was inadequate when rainfall was low, even in your equatorial habitat. What independent evidence do you have that the lactating females couldn't meet their energy needs fully even when rainfall was low?

our data strongly suggests that rainfall over the prior 30 day period limits the energy intake of female banded mongooses.

Hopefully your new study will provide the data, because if food is sufficient at low rainfall, it is difficult to justify your conclusions.

If food is sufficient for maximal lactation at low rainfall, it would be difficult to explain the reduction in pup mass as rainfall decreases. Our previous studies on the same study system demonstrate the impact of rainfall on mass gain and a variety of behaviours, consistent with rainfall impacting food availability and limiting energy intake.

19. Line 113: were there any heatwaves in your study period?

changed the term 'heatwave' for 'hot conditions'

I would prefer 'hotter conditions'. Across the spectrum of environments to which diurnal animals are exposed, 32C air temperature isn't particularly hot.

We have changed to 'hotter conditions' on L25, L154 and L171.

20. Line 115: more abundant at higher rainfall doesn't necessarily mean inadequate at lower rainfall.

This comment has been addressed in the previous round of review.

21.Line 117: what other variables did the weather station measure? You have not reported any. Did you have access to any measure of radiation, water vapour pressure or wind speed? Convective heat loss could have been higher at the higher air temperatures than at the lower ones if wind speed increased sufficiently at times when air temperature was high. So, depending on the wind, lactating females could have dissipated heat to the air more easily at the higher air temperatures.

This comment has been addressed in the previous round of review.

22.Line 120-122: describe your “methods” in the past tense. You are telling us what you have done, not what you are doing.

This comment has been addressed in the previous round of review.

23.Line 128: when the pups were caught by hand, you had an ideal opportunity to measure their rectal temperatures, which would have allowed you to test pup body temperature as a covariate, and to eliminate any febrile pups, which was important because fever induces anorexia.

In order to avoid dishabituating the animals and causing undue stress, we do not manipulate the pups during handling,

I think that the major stress would have come from the act of catching the pups. If their genitals were exposed, so presumably was the anus, and I think that you could have measured temperature within 30 s with a fine-wire thermocouple. Those data would have been unique and invaluable. You also could have obtained a surface temperature of the underbelly instantaneously with an infrared thermometer.

As we previously explained, we did not have the permits to take anal temperatures from pups without anaesthesia, and were concerned about dishabituation. Our field team has over 20 years of experience with our study species, and we follow their advice regarding handling procedures.

24.Line 132: if the balance accuracy was 1.5 g, how could you measure mass to a tenth of a gram and its SE to a hundredth of a gram? Also, SD not SE is the appropriate statistic here, because you are concerned with variance within the population not between populations.

25.Line 146: what was the accuracy of the thermometer in the weather station? Giving air temperatures to the second decimal place implies that the thermometers were accurate to 1/100°C. In the supplemental information, you gave some temperatures to eight decimal places, when your thermometer probably wasn't accurate even to one decimal place.

Raw temperature data were measured to 0.1 degree, which we have now explained in the main text (L111). Here, we present mean values rather than raw data, and there is no requirement for summary statistics to be presented at the same precision as the data. Nevertheless, we have now changed all values in the main text to 1dp at your suggestion. In the dataframe, we retain mean values as they are, as reducing the number of decimal places will not qualitatively influence our results.

Your raw temperature was measured to a resolution of 0.1 degree, not an accuracy of 0.1 degree; you do not say that you calibrated the thermometer. Nevertheless, you are correct that, with sufficient measurements, the mean can be more precise than individual measurements.

I'll leave the decision about the supplemental information to the editor. I wouldn't like my journal to be cluttered with meaningless numbers.

We will change the supplementary data that we will upload to Dryad should the editor prefer. However, we do not see the need to reduce the resolution of data used in our study as the storage requirements for the extra digits are minimal.

26.Line 146: again, giving rainfall to two decimal places implies that the rain gauges were accurate to 1/100 mm.

This comment has been addressed in the previous round of review.

27.Line 146-149: I think that the appropriate covariate would have been ratio of number of pups to number of lactating females.

We agree that this would be a more appropriate covariate. However, as pups are born underground and remain there for the first ~30 days of their life, we are unable to count the number of pups at birth, and it is likely that some pups die before we are able to observe them. Due to the resulting uncertainty in pup:female ratios at birth, we therefore include in our model the number of lactating females.

I had in mind ratio of lactating females to pups at emergence, which I think is a more appropriate covariant than raw number of lactating females, though what you really would like is the ratio at birth.

Please see our previous justification for the measure we used (above).

28.Line 175, Figure 1: were the slopes of the regression lines significantly different to zero? If they were, why would pup mass decrease with increasing prior rainfall at the highest temperature? According to your postulates, that trend implies less maternal food intake at the higher rainfall. Why? Or is there another factor at play?

This comment has been addressed in the previous round of review.

29.Line 175, Figure 1: though I think that air temperature was relatively unimportant both in determining heat load (radiation would have much more important) on the lactating females, and in determining their capacity to dissipate body heat (water vapour pressure would have been much more important), if you want to make an argument based on air temperature, you need to show that the sets of temperatures represented by your three means were different statistically.

This comment has been addressed in the previous round of review.

30.Line 176: again, implying that you could measure air temperature to an accuracy of 0.01°C.

This comment has been addressed in the previous round of review.

31.Line 181: only if the thermoregulation of the lactating animals was compromised by the higher air temperature. We don't know what their surface temperature was (it can be measured remotely by thermography) but it would have been high when they were in the sun, perhaps even higher than deep-body temperature. Let's say 40°C. Then the gradient for convective heat loss was $40-28.3 = 11.7^{\circ}\text{C}$ at your lowest temperature and $40-30.9 = 9.1^{\circ}\text{C}$ at your highest temperature. So convective heat loss would have differed by only about 20%, and that assumes that wind speed was the same at all air temperatures. Even if the animals weren't cooling by evaporation, that's a small difference. To be consistent with HDL theory, you would have to say that the lactating females were "forced to suppress exothermic processes" (line 183, i.e. to reduce their lactation sufficiently to prevent the pups growing) because they "knew" that their cooling by convection was going to be reduced by 20%, an amount easily compensable by evaporative cooling.

the HDL hypothesis is not about predicting future conditions.

I think that it is, and so, I believe, do Speakman and Król.

Our reading of Speakman and Krol's work does not bring us to the same conclusion. For example, in their 2010 review (*Journal of Animal Ecology*, 79, 726–746) in which they discuss the origin of the hypothesis, they acknowledge that previous work has found that lactation is reduced at higher temperatures, and they do not distinguish their hypothesis based on predicting future conditions. "In fact, the role of heat as a factor constraining mammalian reproduction had been known for some considerable time before our 'discovery' in lactating mice. The context, however, of these previous studies was different and concerned reproductive performance of domesticated livestock (e.g. Ominski et al. 2002; Lacetera et al. 2003; Renaudeau, Noblet & Dourmad 2003; Odongo et al. 2006). The novelty in our work was to discover that a factor of key importance constraining the lactation performance of large 50–500 kg animals like sheep, pigs and cattle, with low surface to volume ratios, was also important in mice weighing only 30–40 g with a surface to volume ratio 20–30 times greater. The commonality of this constraint across such a large range of body masses suggested the potential for heat dissipation capacity to be a fundamentally unifying factor in our understanding of energetic constraints affecting endotherms."

our findings are consistent with HDL theory (rather than stating that they are a result of HDL).

More accurately stated, I think.

32.Line 189-190: so, you are saying that air temperatures reaching a maximum of 30.9°C outside the den might have imposed a risk of harmful hyperthermia on pups in the den. In adult slender mongoose in the lab, rectal temperature was elevated but held constant when the air temperature was 38°C permanently (Kamau et al. 1979).

In both cases the authors specifically state that body (rectal) temperatures continued increasing at higher temperatures, in contrast to the reviewer's claim.

I said at 38C, not 43C

We already explain why it is unlikely that pups are experiencing hyperthermia in the den (L185-192).

33.Line 192: there would have been benefit only when the pups were in the den. As soon as they were outside, and under radiant heat load, a high surface area to mass ratio would have been a disadvantage. The larger pups would have been better off outside the den.

This comment has been addressed in the previous round of review.

34.Line 199: Van de Ven et al. 2020 is relevant to this paragraph.

This comment has been addressed in the previous round of review.

35.Line 209: observing pups in the den may have been beyond your resources, but It certainly was possible, with fibre-optic cameras.

Over the 20+ years of our study, we have indeed tried to observe pups in the den using such cameras, however due to the positioning of dens (often in inaccessible locations such as dense bushes or underneath buildings), we have been unable to reliably collect such observations.

I can imagine how frustrating that must have been because you could have added so much more by seeing the pups.

36.Line 212: I don't think that the comparisons discussed in this paragraph are relevant. Your pups did not experience high temperatures in early ex-utero development.

This comment has been addressed in the previous round of review.

37.Line 222-224: your analysis did not confirm an effect of temperature, but only of a temperature/rainfall interaction.

By definition, a variable involved in an interaction has an effect.

You are correct. Just not an independent effect, although your subsequent analysis indeed did reveal an independent effect.

38.Line 224: I do not understand why would you expect high food abundance to compensate for high ambient heat load? High food abundance presumable would lead to more milk consumption and higher metabolic heat production. If you are working within HDL theory, how does that compensate for high ambient heat load?

This comment has been addressed in the previous round of review.